# Spectral Imbalance Causes Forgetting in Low-Rank Continual Adaptation

Hao Gu [* 1 2]  Mao-Lin Luo [* 1 2]  Zi-Hao Zhou [1 2]  Han-Chen Zhang [1 2]  Min-Ling Zhang [1 2]  Tong Wei [1 2]

## Abstract

Parameter-efficient continual learning aims to adapt pre-trained models to sequential tasks without forgetting previously acquired knowledge. Most existing approaches treat continual learning as avoiding interference with past updates, rather than considering what properties make the current task-specific update naturally preserve previously acquired knowledge. From a knowledge-decomposition perspective, we observe that low-rank adaptations exhibit highly imbalanced singular value spectra: a few dominant components absorb most of the adaptation energy, thereby (i) more likely to disrupt previously acquired knowledge and (ii) making the update more vulnerable to interference from subsequent tasks. To enable explicit balance among components, we decouple the *magnitude* of the task update from its *directional structure* and formulate it as a constrained optimization problem on a restricted Stiefel manifold. We address this problem using a projected first-order method compatible with standard deep-learning optimizers used in vision-language models. Our method mitigates both backward and forward forgetting, consistently outperforming continual learning baselines. The implementation code is available at https://github.com/haodotgu/EBLoRA.

## 1. Introduction

Vision-language models (VLMs) have demonstrated remarkable capabilities in a wide range of tasks, making them cornerstones of many downstream applications (Comanici

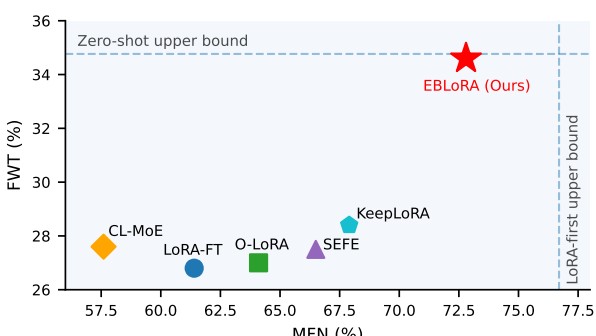

*Figure 1.* Comparison of parameter-efficient methods on the UCIT benchmark in terms of MFN and FWT. MFN measures the model's final performance after learning all tasks, whereas FWT reflects the ability to generalize learned knowledge to unseen tasks. Zero-shot upper bound is the mean accuracy of the base model, while LoRA-first upper bound is the mean accuracy of LoRA on each task immediately after it is learned. EBLoRA clearly outperforms all baselines in MFN and FWT, achieving performance close to both upper bounds.

et al., 2025; Achiam et al., 2023; Radford et al., 2021). As real-world data evolves over time, there is an increasing need for these models to adapt to new knowledge, motivating continual learning (CL). CL methods aim to train a single model on a sequence of tasks while preserving both the pre-trained knowledge and the knowledge acquired from earlier tasks (Mukhoti et al., 2024; Zheng et al., 2023). In a typical setting, the model is trained on tasks that arrive sequentially with limited or no access to past data, making it challenging to prevent interference when adapting to new tasks (Parisi et al., 2019; Zhou et al., 2024). The interference can corrupt learned knowledge and substantially degrade the performance of previous tasks (Guo et al., 2025b).

To mitigate interference, existing works typically modify the optimization process or allocate parameters at the task level (Kirkpatrick et al., 2017; Zhang et al., 2024; Luo et al., 2025). A common line of methods introduces replay buffers to approximate joint training by revisiting stored or generated past samples (Wu et al., 2025; Zheng et al., 2023). Regularization-based methods constrain parameter updates to preserve parameters important for previous tasks, e.g., via parameter-importance penalties or encouraging gradient directions across tasks to be orthogonal (Flesch et al.,

---

*Equal contribution

This work was done when Hao Gu was interning at SEU. [1]School of Computer Science and Engineering, Southeast University, Nanjing 210096, China [2]Key Laboratory of Computer Network and Information Integration (Southeast University), Ministry of Education, China. Correspondence to: Min-Ling Zhang <zhangml@seu.edu.cn>, Tong Wei <weit@seu.edu.cn>.

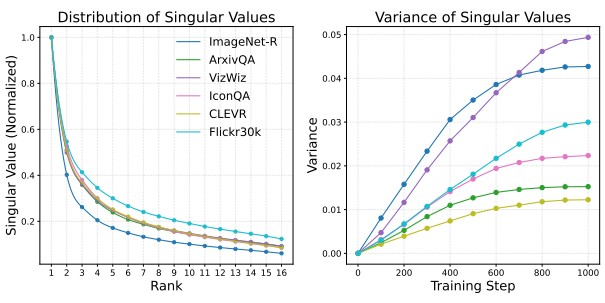
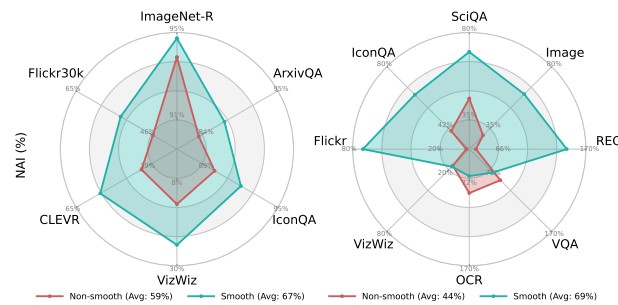

*(a)* Singular value distribution and its variance over training

*(b)* Multi-task merging performance (NAI) w/ vs. w/o smoothing

*Figure 2.* Imbalanced low-rank components amplify cross-task interference. (a) LoRA updates exhibit long-tailed singular value spectra where a few components dominate adaptation energy, while variance increases during training. (b) Multi-task LoRA merging reveals that direct merging degrades performance, while singular value smoothing can reduce interference and improve task coexistence.

2022; Xie et al., 2022). Architecture-based methods allocate task-specific capacity, such as branches or experts (Yu et al., 2024; Dou et al., 2024), while sharing a common backbone to avoid conflicts between tasks (Li et al., 2025a).

Prior works primarily focus on preventing interference with previously acquired knowledge, yet largely overlook the nature of the new task update itself. This raises a fundamental question: *what internal properties of an update are inherently beneficial to continual learning?*

In this work, we revisit continual learning from a *knowledge-component* perspective. Inspired by recent advances in low-rank adaptation (LoRA) (Hu et al., 2022), we view task-specific updates as low-rank modifications to a pre-trained model. Specifically, by employing singular value decomposition, the update of the $t$-th task $\Delta \mathbf{W}_t$ can be decomposed as a set of rank-one components $\{\sigma_{t,i} \mathbf{u}_{t,i} \mathbf{v}_{t,i}^\top\}$, where each component encodes a distinct input–output interaction pattern. From this viewpoint, learning a new task corresponds to adding and reshaping these knowledge components in parameter space. Empirically, we observe that the singular value spectra of such updates exhibit a pronounced imbalance: a few directions dominate most of the adaptation energy, while many weaker yet semantically useful directions are suppressed. Such spectral imbalance causes the model to rely excessively on a small set of directions, which in turn reduces cross-task generalization and makes these directions disproportionately vulnerable to interference from subsequent tasks. These findings suggest that catastrophic forgetting is not merely an optimization issue, but a structural consequence of imbalanced competition among knowledge components.

Building on this insight, we argue that effective continual learning should explicitly regulate the internal properties of task-specific updates. Instead of treating $\Delta \mathbf{W}_t$ as an unconstrained low-rank matrix, we propose to decouple the *magnitude* and the *directional structure* of task updates by factorizing $\Delta \mathbf{W}_t = s_t \mathbf{U}_t \mathbf{V}_t^\top$, where $s_t$ controls the global

adaptation energy and the columns of $\mathbf{U}_t$ and $\mathbf{V}_t$ form orthonormal bases. This formulation allows us to explicitly maintain *balance among components* and to further constrain the update directions to remain orthogonal to previous tasks, reducing interference during continual learning.

To train the model under this parameterization, we formulate the learning problem on a restricted Stiefel manifold and solve it by employing a projected optimization method that seamlessly integrates with standard deep-learning training pipelines, enforcing the structural constraints with minimal modification to existing models and training workflows. Fig. 1 compares our approach with parameter-efficient continual learning methods in terms of MFN and FWT. Our approach substantially mitigates forgetting of knowledge from both pre-training and previously fine-tuned tasks, and consistently achieves better continual learning performance than previous approaches.

In summary, our main contributions are threefold:

- We introduce a knowledge-component perspective of LoRA weights, showing that interference arises from spectral imbalance among knowledge components in task-specific updates.

- This key finding motivates a new parameter factorization that decouples the magnitude from the directional structure of updates, explicitly imposing a balanced learning of knowledge components.

- We formulate the problem as restricted Stiefel manifold optimization and propose a new continual learning approach called EBLoRA. Both theoretical and empirical analyses justify the effectiveness of our approach.

## 2. Motivation

To motivate our approach, this section examines the spectral structure of low-rank adaptations and how such a structure affects continual learning. In Sec. 2.1, we introduce

a knowledge-component view in which LoRA updates are decomposed into rank-one components. In Sec. 2.2, we present empirical evidence that these components exhibit a highly imbalanced singular value distribution, and show that such imbalance amplifies cross-task interference, motivating the need for learning balanced knowledge components.

## 2.1. Knowledge Components in Continual Learning

In this part, we revisit continual learning from a *knowledge-component* perspective. Following recent advances in low-rank adaptation, we view parameter-efficient fine-tuning methods (e.g., LoRA) as a structured form of task updates. Given a pre-trained model with parameters $\mathbf{W}_0$, the update induced by task $t$ can be written as a low-rank matrix $\Delta \mathbf{W}_t = \mathbf{W}_t - \mathbf{W}_{t-1}$, which is explicitly parameterized as $\Delta \mathbf{W}_t = \mathbf{B}_t \mathbf{A}_t$ in the LoRA setting. Such updates admit a compact singular value decomposition as follows:

$$\Delta \mathbf{W}_t = \sum_{i=1}^{r_t} \sigma_{t,i}\, \mathbf{u}_{t,i} \mathbf{v}_{t,i}^\top, \tag{1}$$

where each outer product $\mathbf{u}_{t,i}\mathbf{v}_{t,i}^\top$ defines a rank-one update direction in the parameter space. Functionally, $\mathbf{u}_{t,i}$ specifies an input-side feature direction and $\mathbf{v}_{t,i}$ determines how this feature is mapped to the output. The $(\mathbf{u}_{t,i}, \mathbf{v}_{t,i})$ pair therefore determines the *directional structure* of the update, while the associated singular value $\sigma_{t,i}$ controls its *magnitude* (i.e., adaptation energy) along that direction.

This decomposition offers a new perspective on continual learning: *sequential training can be viewed as the accumulation and modification of knowledge components.*

## 2.2. Observation: Imbalance among Components Amplifies Interference

A notable yet underexplored observation is that LoRA updates exhibit a highly long-tailed singular value distribution, where most of the adaptation energy is concentrated in a few dominant components. Fig. 2a (left) shows the normalized singular values of LoRA trained on UCIT tasks, revealing that the first few ranks capture the majority of the spectral energy. This suggests that LoRA allocates adaptation energy in an imbalanced manner: a small set of directions is amplified excessively, whereas many potentially useful ones remain weakly expressed. Moreover, as shown in Fig. 2a (right), the variance of singular values increases throughout training, indicating that this imbalance is progressively reinforced during the optimization process.

Such imbalance has important implications for continual learning. We argue that the highly imbalanced updates are (i) more disruptive to previously acquired knowledge, and (ii) more vulnerable to interference from subsequent tasks, since most of the adaptation energy is concentrated in very

few dominant components. Both effects increase cross-task interference during continual learning.

To examine this hypothesis, we conduct a controlled LoRA merging (Zhang et al., 2026) experiment that simulates the accumulation of task adaptations. For each benchmark, we independently fine-tune one LoRA adapter per task and then merge them to form a multi-task model. Since merging aggregates updates without further optimization, performance degradation directly reflects interference between the task-specific updates.

In Fig. 2b, we consider two variants: direct merging (*Non-smooth*) (Ilharco et al., 2023) and a smoothed version (*Smooth*). In the latter case, for each task update $\Delta \mathbf{W}_t$ with singular values $\boldsymbol{\sigma}_t = (\sigma_{t,1}, \dots, \sigma_{t,r})$ sorted in descending order, we form a *balanced* variant by replacing $\boldsymbol{\sigma}_t$ with a constant vector $\bar{\boldsymbol{\sigma}}_t = (\frac{1}{r}\sum_{j=1}^{r} \sigma_{t,j})\mathbf{1}_r$, while keeping the singular vectors unchanged.

We quantify how much of the knowledge gained through task-specific fine-tuning is preserved after merging using the *Normalized Accuracy Improvement* (NAI) (Marczak et al., 2025) defined as:

$$\frac{A_{\text{merged}} - A_{\text{zero-shot}}}{A_{\text{individual}} - A_{\text{zero-shot}}}, \tag{2}$$

where $A_{\text{merged}}$ denotes the accuracy on the target task after merging the LoRA adapters from all tasks, $A_{\text{zero-shot}}$ denotes the accuracy of the base model on this task, and $A_{\text{individual}}$ denotes the accuracy obtained by individually fine-tuning on that task. Empirically, we find that direct merging leads to substantial drops in NAI, whereas applying singular value smoothing prior to merging significantly mitigates the drop, suggesting that the imbalanced singular value spectra contribute to cross-task interference.

The observation above suggests that *one should explicitly regulate balance among components to stabilize low-rank adaptation across tasks*, motivating our approach in the following section.

## 3. The Proposed Approach

This section presents our method, *Energy-Balanced Low-Rank Adaptation* (EBLoRA). In Sec. 3.1, we propose a structured parameterization of task updates that balances adaptation energy and incorporates gradient-orthogonality constraints, leading to a constrained optimization problem. In Sec. 3.2, we present an efficient optimization algorithm that solves the constrained problem using projected updates on a restricted Stiefel manifold. In Sec. 3.3, we analyze the theoretical properties of the proposed optimization method, showing that both the projection and retraction steps are optimal under Riemann geometric constraints.

### 3.1. Restricted Stiefel Optimization Problem

**Balanced task updates.** Instead of treating $\Delta\mathbf{W}_t$ as an indivisible parameter change, we argue that continual learning should regulate how adaptation energy is distributed across its underlying knowledge components. To this end, we factorize each task update as follows.

$$\Delta\mathbf{W}_t = s_t\,\mathbf{U}_t\mathbf{V}_t^\top, \quad (3)$$

where $s_t \in \mathbb{R}$ is a scalar that controls the overall magnitude of task update, and $\mathbf{U}_t, \mathbf{V}_t$ are matrices whose columns form orthonormal bases. This factorization decouples the *magnitude* of the task update from its *directional structure*, enabling us to explicitly reason about and regulate the balance of internal knowledge components.

Building on this formulation, we introduce a learning framework that stabilizes task adaptation by maintaining balanced and well-conditioned knowledge components across tasks.

**Mitigating interference via gradient orthogonality.** To further mitigate backward forgetting, we explicitly constrain the direction basis of the current task update to avoid interfering with directions that are important for previous tasks. Let $\mathbf{G}_{t-1} \in \mathbb{R}^{d \times k}$ denote a matrix whose columns store a set of representative gradient directions accumulated from tasks $\{1, \ldots, t-1\}$ (e.g., sampled low-rank summaries of per-layer gradient as GPM (Saha et al., 2021)). We enforce the current update basis $\mathbf{U}_t$ to be orthogonal to the subspace spanned by $\mathbf{G}_{t-1}$:

$$\mathbf{G}_{t-1}^\top \mathbf{U}_t = \mathbf{0}. \quad (4)$$

Intuitively, this constraint prevents the new task from allocating its update directions along previously sensitive gradient modes, thereby reducing interference.

**Restricted Stiefel manifold.** Recall that our factorized task update $\Delta\mathbf{W}_t = s_t\mathbf{U}_t\mathbf{V}_t^\top$ with $\mathbf{U}_t, \mathbf{V}_t^\top$ are orthonormal bases. We impose the standard Stiefel constraints:

$$\mathbf{U}_t^\top \mathbf{U}_t = \mathbf{I}_r, \quad \mathbf{V}_t^\top \mathbf{V}_t = \mathbf{I}_r, \quad (5)$$

together with the gradient-orthogonality constraint in Eq. (4). This defines a *restricted Stiefel manifold* for $\mathbf{U}_t$:

$$\mathcal{M}_t = \left\{ \mathbf{U} \in \mathbb{R}^{d \times r} \mid \mathbf{U}^\top \mathbf{U} = \mathbf{I}_r,\ \mathbf{G}_{t-1}^\top \mathbf{U} = \mathbf{0} \right\}. \quad (6)$$

Accordingly, the learning problem for task $t$ becomes a constrained optimization on $\mathbb{R} \times \mathcal{M}_t \times \mathrm{St}(d, r)$:

$$\min_{s_t \in \mathbb{R},\ \mathbf{U}_t \in \mathcal{M}_t,\ \mathbf{V}_t \in \mathrm{St}(d,r)} \mathcal{L}_t\big(W_{t-1} + s_t\mathbf{U}_t\mathbf{V}_t^\top\big) \quad (7)$$

where $\mathrm{St}(d, r) = \{\mathbf{X} \in \mathbb{R}^{d \times r} \mid \mathbf{X}^\top \mathbf{X} = \mathbf{I}_r\}$.

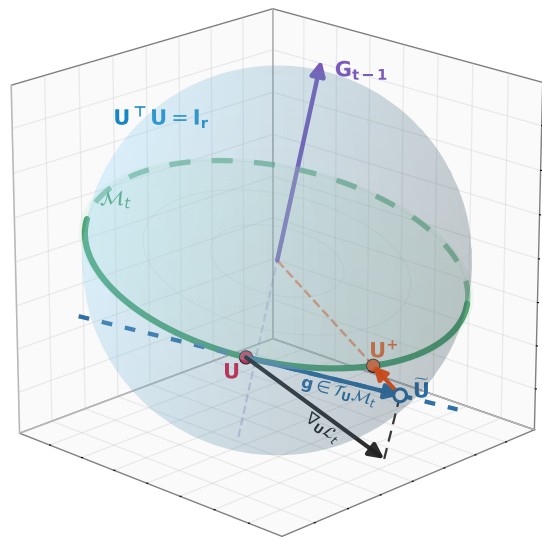

*Figure 3.* An intuitive geometric illustration of one-step optimization (Alg. 1) on the restricted Stiefel manifold $\mathcal{M}_t$ in low dimensions. The feasible set $\mathcal{M}_t$ is the intersection between the unit sphere $\mathbf{U}^\top \mathbf{U} = \mathbf{I}_r$ and the linear constraint subspace $\mathrm{Null}(\mathbf{G}_{t-1}^\top)$. Starting from the current iterate $\mathbf{U} \in \mathcal{M}_t$, the Euclidean gradient $\nabla_{\mathbf{U}}\mathcal{L}_t$ is first projected onto the tangent space $\mathcal{T}_{\mathbf{U}}\mathcal{M}_t$ to obtain the Riemannian gradient $\mathbf{g}$. A tangent-space update yields the tentative point $\widetilde{\mathbf{U}}$, which is then retracted back to the manifold to produce the next feasible iterate $\mathbf{U}^+ \in \mathcal{M}_t$.

### 3.2. Optimization over the restricted Stiefel manifold

Before detailing the individual components, we briefly summarize our Riemannian optimization procedure. Since the update basis $\mathbf{U}_t$ is constrained to lie on a manifold feasible set, standard Euclidean optimization is no longer applicable. Instead, optimization is performed by alternating between two conceptually simple steps: (i) mapping unconstrained updates to the tangent space of the constraint manifold, where valid descent directions are defined; and (ii) transporting the updated parameters back onto the manifold to restore feasibility. This framework allows us to leverage standard first-order optimizers while explicitly respecting the geometric structure induced by the orthogonality and gradients subspace constraints.

**Tangent-space projection.** Recall that the update basis $\mathbf{U}_t$ is constrained to lie on the restricted manifold $\mathcal{M}_t = \{\mathbf{U} \mid \mathbf{U}^\top \mathbf{U} = \mathbf{I},\ \mathbf{G}_{t-1}^\top \mathbf{U} = \mathbf{0}\}$, which is the intersection of the Stiefel manifold and a linear subspace constraint. Accordingly, the tangent space at $\mathbf{U}$ is given by

$$\mathcal{T}_{\mathbf{U}}\mathcal{M}_t = \left\{ \mathbf{Z} \in \mathbb{R}^{d \times r} \mid \mathbf{U}^\top \mathbf{Z} + \mathbf{Z}^\top \mathbf{U} = \mathbf{0},\ \mathbf{G}_{t-1}^\top \mathbf{Z} = \mathbf{0} \right\}. \quad (8)$$

To project an ambient matrix $\mathbf{Z}$ onto this tangent space, we remove the component that violates the linear constraint:

$$\mathbf{Z}_0 = \mathbf{P}_{\perp \mathbf{G}_{t-1}} \mathbf{Z}, \qquad (9)$$

where $\mathbf{P}_{\perp \mathbf{G}_{t-1}} = \mathbf{I} - \mathbf{G}_{t-1} \mathbf{G}_{t-1}^\top$ denotes the projector onto the null space of $\mathbf{G}^\top$. Then, we apply the standard Stiefel tangent projection within the resulting subspace:

$$\mathcal{P}_{\mathcal{T}_{\mathbf{U}} \mathcal{M}_t}(\mathbf{Z}) = \mathbf{Z}_0 - \mathbf{U}_t \operatorname{sym}(\mathbf{U}^\top \mathbf{Z}_0), \qquad (10)$$

where $\operatorname{sym}(\mathbf{A}) = \frac{1}{2}(\mathbf{A} + \mathbf{A}^\top)$. Since both $\mathbf{U}_t$ and $\mathbf{Z}_0$ lie in the null space of $\mathbf{G}_{t-1}^\top$, the resulting projection automatically satisfies $\mathbf{G}_{t-1}^\top \mathcal{P}_{\mathcal{T}_{\mathbf{U}} \mathcal{M}_t}(\mathbf{Z}) = \mathbf{0}$ and therefore lies in the valid tangent space.

**Manifold retraction.** Given a tentative update $\widetilde{\mathbf{U}}$, we retract it back to $\mathcal{M}_t$ in two steps: we first remove the $\mathbf{G}$-component and then enforce orthonormality by the whitening operator. Concretely, define

$$\mathbf{Y} = \mathbf{P}_{\perp \mathbf{G}_{t-1}} \widetilde{\mathbf{U}}, \qquad (11)$$

and then apply the whitening retraction (Koivunen & Kostinski, 1999) as follows:

$$\mathbf{U}^+ = \operatorname{whiten}(\mathbf{Y}) \triangleq \mathbf{Y}(\mathbf{Y}^\top \mathbf{Y})^{-\frac{1}{2}}. \qquad (12)$$

This operation ensures $\mathbf{U}^{+\top} \mathbf{U}^+ = \mathbf{I}_r$, and since $\mathbf{Y} \in \operatorname{Null}(\mathbf{G}^\top)$, it also preserves the linear constraint $\mathbf{G}^\top \mathbf{U}^+ = \mathbf{0}$. In practice, $(\mathbf{Y}^\top \mathbf{Y})^{-\frac{1}{2}}$ can be computed via an eigendecomposition (or SVD) of the small $r \times r$ matrix $\mathbf{Y}^\top \mathbf{Y}$.

**Practical implementation with standard optimizers.** Although the formulation suggests Riemannian optimization, classical Riemannian solvers are impractical in deep learning and can be incompatible with standard first-order optimizers such as SGD with momentum or Adam. We therefore adopt a projected optimization scheme that combines manifold constraints with standard optimizers as in Alg. 1: Euclidean gradients are projected onto the tangent space (line 3), updated by the optimizer (line 6), projected again to remove infeasible components (line 8), and finally retracted onto the manifold (line 13). Fig. 3 provides a geometric illustration of one-step optimization on the restricted Stiefel manifold in low dimensions, where the standard Stiefel manifold is represented by the unit sphere in three-dimensional space. This retains the behavior of modern optimizers while enforcing the constraints that help reduce cross-task interference.

**Initialization of structured task updates.** For the $t$-th task, we initialize $(s_t, \mathbf{U}_t, \mathbf{V}_t)$ from the projected task gradient. We first collect a small gradient snapshot $\mathcal{G}_t$ from task $t$ and project it onto the null space of $\mathbf{G}_{t-1}$ via

---

**Algorithm 1** One-step Optimization for $\mathbf{U}$ on $\mathcal{M}_t$

**Require:** Current $\mathbf{U} \in \mathcal{M}_t$ with $\mathbf{U}^\top \mathbf{U} = \mathbf{I}_r$ and $\mathbf{G}^\top \mathbf{U} = \mathbf{0}$; stored directions $\mathbf{G} \in \mathbb{R}^{d \times k}$; Euclidean gradient $\nabla_{\mathbf{U}} \mathcal{L}_t$; optimizer state $\xi_U$.

**Ensure:** Updated $\mathbf{U}^+ \in \mathcal{M}_t$ and optimizer state $\xi_U^+$.

1: $\mathbf{P}_{\perp \mathbf{G}} \leftarrow \mathbf{I} - \mathbf{G} \mathbf{G}^\top$
2: $\operatorname{sym}(\mathbf{A}) \leftarrow \frac{1}{2}(\mathbf{A} + \mathbf{A}^\top)$
3: **// Pre-step tangent projection of the gradient**
4: $\mathbf{Z} \leftarrow \mathbf{P}_{\perp \mathbf{G}} \nabla_{\mathbf{U}} \mathcal{L}_t$
5: $\mathbf{g} \leftarrow \mathbf{Z} - \mathbf{U} \operatorname{sym}(\mathbf{U}^\top \mathbf{Z})$     $\{\mathbf{g} \in \mathcal{T}_{\mathbf{U}} \mathcal{M}_t\}$
6: **// Optimizer step in the tangent space**
7: $(\Delta, \xi_U^+) \leftarrow \text{OptStep}(\mathbf{g}, \xi_U)$     $\{$e.g., SGD-momentum/Adam produces an increment $\Delta\}$
8: **// Post-step tangent correction of the increment**
9: $\mathbf{Z}_\Delta \leftarrow \mathbf{P}_{\perp \mathbf{G}} \Delta$
10: $\Delta \leftarrow \mathbf{Z}_\Delta - \mathbf{U} \operatorname{sym}(\mathbf{U}^\top \mathbf{Z}_\Delta)$     $\{\Delta \in \mathcal{T}_{\mathbf{U}} \mathcal{M}_t\}$
11: **// Apply the corrected increment**
12: $\widetilde{\mathbf{U}} \leftarrow \mathbf{U} + \Delta$
13: **// Manifold retraction via whitening**
14: $\mathbf{Y} \leftarrow \mathbf{P}_{\perp \mathbf{G}} \widetilde{\mathbf{U}}$
15: $\mathbf{U}^+ \leftarrow \mathbf{Y}(\mathbf{Y}^\top \mathbf{Y})^{-\frac{1}{2}}$     $\{\mathbf{U}^+ \in \mathcal{M}_t\}$

---

$\mathcal{G}_t^{\text{proj}} = \mathbf{P}_{\perp \mathbf{G}_{t-1}} \mathcal{G}_t$, which removes components aligned with previously acquired knowledge. SVD of $\mathcal{G}_t^{\text{proj}}$ then provides a task-aligned basis, from which we take the leading $r$ singular vectors to form $\mathbf{U}_t^{(0)}$ and $\mathbf{V}_t^{(0)}$. By construction, $\mathbf{U}_t^{(0)}$ already satisfies $\mathbf{G}_{t-1}^\top \mathbf{U}_t^{(0)} = 0$, ensuring a feasible initialization that respects the gradient-orthogonality constraint. The scalar $s_t$, which controls the overall update magnitude, is initialized separately. Recent work such as LiNeS (Wang et al., 2025) show that deeper layers benefit from larger update scales, while shallower layers prefer smaller ones. Following this trend, we adopt a simple depth-aware initialization for $s_t$, which stabilizes optimization in practice. Detailed information of depth-aware initialization is presented in Appendix B.1 and the complete training procedure is summarized in Alg. 2.

### 3.3. Theoretical Analysis

To complete our projected optimization formulation, it is necessary to verify that the projection introduced above satisfies two key geometric properties (Edelman et al., 1999): (i) the tangent-space projection (Eqs. 9,10) must produce valid descent directions on the restricted manifold; and (ii) the retraction step (Eqs. 11,12) must map tentative iterates to the closest feasible point. These two properties are formally stated below.

**Proposition 3.1** (Optimality of the tangent-space projection). *Let* $\mathcal{M}_t = \{\mathbf{U} \in \mathbb{R}^{d \times r} \mid \mathbf{U}^\top \mathbf{U} = \mathbf{I}, \ \mathbf{G}^\top \mathbf{U} = \mathbf{0}\}$, *and let* $\mathbf{U} \in \mathcal{M}_t$ *be a feasible point. For any ambient matrix*

$\mathbf{Z} \in \mathbb{R}^{d \times r}$, *define:*

$$\mathcal{P}_{\mathcal{T}_{\mathbf{U}} \mathcal{M}_t}(\mathbf{Z}) = \mathbf{Z}_0 - \mathbf{U} \operatorname{sym}(\mathbf{U}^\top \mathbf{Z}_0), \quad \mathbf{Z}_0 = (\mathbf{I} - \mathbf{G}\mathbf{G}^\top)\mathbf{Z}. \tag{13}$$

*Then* $\mathcal{P}_{\mathcal{T}_{\mathbf{U}} \mathcal{M}_t}(\mathbf{Z})$ *is the unique solution of*

$$\min_{\mathbf{Y} \in \mathcal{T}_{\mathbf{U}} \mathcal{M}_t} \quad \|\mathbf{Y} - \mathbf{Z}\|_F^2, \tag{14}$$

*and hence the orthogonal projection of* $\mathbf{Z}$ *onto the tangent space* $\mathcal{T}_{\mathbf{U}} \mathcal{M}_t$ *under the Frobenius norm.*

**Remark.** Proposition 3.1 establishes that the proposed operator computes the unique orthogonal projection onto the tangent space of $\mathcal{M}_t$ under the Frobenius inner product. Consequently, the projected gradient used in Alg. 1 corresponds to the canonical Riemannian gradient on the restricted Stiefel manifold. This guarantees that our projected optimizer performs valid first-order descent while explicitly respecting both orthonormality and gradient-orthogonality constraints. Importantly, it ensures that no optimization signal is removed beyond what is strictly required for feasibility, thereby retaining compatibility with standard deep-learning optimizers such as SGD or Adam.

**Proposition 3.2** (Optimality of the manifold retraction). *Let* $\mathcal{M}_t = \{\mathbf{U} \in \mathbb{R}^{d \times r} \mid \mathbf{U}^\top \mathbf{U} = \mathbf{I}_r, \, \mathbf{G}^\top \mathbf{U} = \mathbf{0}\}$ *and let* $\widetilde{\mathbf{U}} \in \mathbb{R}^{d \times r}$ *be an arbitrary ambient iterate. Consider the two-step retraction*

$$\mathbf{Y} = (\mathbf{I} - \mathbf{G}\mathbf{G}^\top)\widetilde{\mathbf{U}}, \quad \mathbf{U}^+ = \mathbf{Y}(\mathbf{Y}^\top \mathbf{Y})^{-\frac{1}{2}}. \tag{15}$$

*Then* $\mathbf{U}^+ \in \mathcal{M}_t$ *and* $\mathbf{U}^+$ *is the unique solution to*

$$\min_{\mathbf{U} \in \mathcal{M}_t} \quad \|\mathbf{U} - \widetilde{\mathbf{U}}\|_F^2, \tag{16}$$

*i.e., the closest feasible point to* $\widetilde{\mathbf{U}}$ *in the Frobenius norm.*

**Remark.** Proposition 3.2 shows that the proposed retraction is not merely a feasibility-restoring heuristic, but in fact computes the unique closest point to the tentative iterate $\widetilde{\mathbf{U}}$ on the restricted Stiefel manifold under the Frobenius norm. This optimality interpretation ensures that no unnecessary distortion is introduced during the retraction step, and that feasibility is maintained with minimal deviation from the optimizer dynamics. In particular, the result implies that our retraction preserves first-order optimization signals while exactly enforcing both orthogonality and gradient-orthogonality constraints.

## 4. Experiments

We evaluate EBLoRA under the standard continual learning setting in which tasks arrive sequentially without access to the past data. In Sec. 4.1, we compare our method with multiple parameter-efficient continual learning baselines

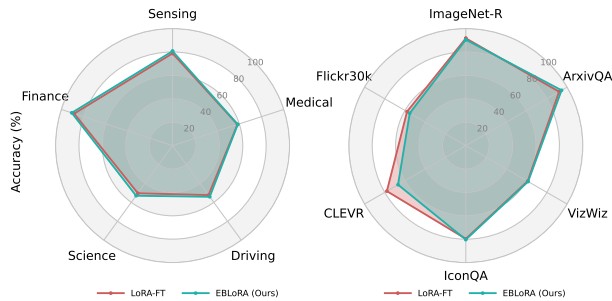

*Figure 4.* Comparison between **LoRA-FT** and **EBLoRA (Ours)** on the **MLLM-DCL** (left) and **UCIT** (right) benchmarks. The radar plots display the accuracy of each task immediately after it is learned, showing that our method preserves the model's ability to continuously learn new tasks.

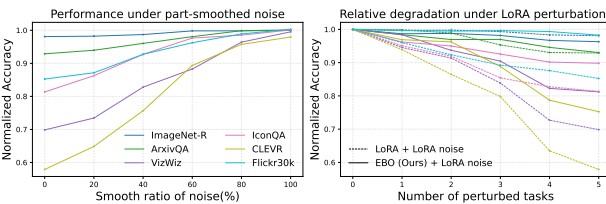

*Figure 5.* Interference analysis of balanced singular value. Noise is generated through merging LoRA weights trained on UCIT tasks. Left: applying partially smoothed noise on LoRA reduces interference in seen tasks. Right: under equal-norm perturbations, EBO approach exhibits higher robustness compared to LoRA.

on two vision-language benchmarks to assess overall performance in realistic multi-task scenarios. In Sec. 4.2, we conduct controlled interference analyses to investigate why spectral balance improves robustness and reduces cross-task interference. Finally, in Sec. 4.3, we present an ablation study that decomposes the contribution of each component of EBLoRA and isolates their effects.

### 4.1. Main Results

We evaluate our method on the encoder-decoder model LLaVA. The experiments (Tab. 1 and 2) are conducted on the UCIT (Guo et al., 2025c) and the MLLM-DCL (Guo et al., 2025c) benchmarks, including various instruction formats such as image captioning, visual question-answer, and multiple-choice questions. Detailed experiment settings are presented in Appendix B.1 and B.2.

We report the accuracy of each learned task after the model has been trained on all tasks. Additionally, we present four aggregated metrics: *Mean Final Accuracy* (MFN), *Mean Average Accuracy* (MAA), *Backward Transfer* (BWT) following SEFE (Chen et al., 2025) and *Forward Transfer* (FWT) following ZSCL (Zheng et al., 2023). This setting stresses both stability and plasticity during sequential adaptation. Definitions of these metrics are provided in Appendix B.3.

EBLoRA achieves state-of-the-art performance on the four aggregated metrics in each of these settings, demonstrating

*Table 1.* Comparison of the proposed EBLoRA method with existing approaches on the UCIT benchmark.

| Method | Accuracy on Each Task (%) | | | | | | Aggregated Results (%) | | | |
|---|---|---|---|---|---|---|---|---|---|---|
| | *ImageNet-R* | *ArxivQA* | *VizWiz* | *IconQA* | *CLEVR* | *Flickr*30k | MFN↑ | MAA↑ | BWT↑ | FWT↑ |
| Zero-shot | 16.3 | 53.7 | 38.4 | 19.2 | 20.6 | 41.9 | | | | |
| LoRA-FT (Hu et al., 2022) | 58.6 | 76.7 | 45.7 | 67.4 | 61.6 | **58.0** | 61.4 | 76.5 | -15.4 | 26.8 |
| O-LoRA (Wang et al., 2023a) | 74.2 | 80.9 | 45.3 | 62.9 | 63.8 | 57.2 | 64.1 | 77.8 | -11.1 | 27.0 |
| CL-MoE (Huai et al., 2025) | 61.2 | 75.8 | 44.4 | 52.6 | 54.4 | 57.3 | 57.6 | 74.4 | -13.9 | 27.6 |
| SEFE (Chen et al., 2025) | 80.2 | 79.1 | 47.1 | 69.4 | 65.7 | 57.3 | 66.5 | 79.1 | -9.8 | 27.5 |
| KeepLoRA (Luo et al., 2026) | 82.4 | 86.7 | 46.6 | 67.8 | **66.4** | 57.2 | 67.9 | 80.1 | -7.9 | 28.4 |
| EBLoRA (Ours) | **89.6** | **94.2** | **56.6** | **75.2** | 66.2 | 55.3 | **72.8** | **82.9** | **-2.0** | **34.6** |

*Table 2.* Comparison of the proposed EBLoRA method with existing approaches on the MLLM-DCL benchmark.

| Method | Accuracy on Each Task (%) | | | | | Aggregated Results (%) | | | |
|---|---|---|---|---|---|---|---|---|---|
| | *Sensing* | *Medical* | *Driving* | *Science* | *Finance* | MFN↑ | MAA↑ | BWT↑ | FWT↑ |
| Zero-shot | 32.3 | 28.3 | 15.6 | 35.6 | 62.6 | | | | |
| LoRA-FT (Hu et al., 2022) | 69.3 | 44.3 | 29.1 | 41.4 | 88.4 | 54.5 | 61.9 | -11.2 | 32.4 |
| O-LoRA (Wang et al., 2023a) | 72.3 | 46.9 | 31.6 | 41.5 | 88.1 | 56.1 | 62.8 | -9.7 | 33.3 |
| CL-MoE (Huai et al., 2025) | 71.8 | 47.4 | 29.5 | 41.5 | 89.2 | 55.9 | 62.4 | -10.5 | 32.6 |
| SEFE (Chen et al., 2025) | 77.1 | 50.9 | 40.3 | 43.0 | 88.4 | 59.9 | 64.6 | -6.0 | 33.5 |
| KeepLoRA (Luo et al., 2026) | 78.8 | 54.3 | 50.2 | 49.5 | 89.3 | 64.4 | 67.3 | -2.1 | 33.7 |
| EBLoRA (Ours) | **79.4** | **57.2** | **53.9** | **52.5** | **90.6** | **66.7** | **68.2** | **-0.7** | **34.4** |

that our approach consistently mitigates backward forgetting and enhances forward transfer.

**Plasticity of EBLoRA.** Importantly, as shown in Fig. 4, EBLoRA retains the ability to continuously acquire new tasks without sacrificing plasticity, indicating that the improved stability does not come at the cost of adaptability.

### 4.2. Why Balanced Energy Helps Continual Learning

To better understand why spectral balance facilitates continual learning, we conduct controlled interference experiments based on LoRA adapters trained on UCIT tasks (Fig. 5). In addition to comparing with standard LoRA, we include a stripped-down variant of our method, referred to as *Energy-Balanced Optimization* (EBO), which applies only the $s\mathbf{UV}$ factorization and energy-balancing mechanism while removing the gradient-orthogonality constraint introduced by Eq. 4 and the depth-aware initialization described in Sec. 3. This ablation isolates the effect of spectral balance itself and allows us to analyze its impact independently of other components. In these experiments, each task-specific adapter is treated as a source of noise to other tasks, allowing us to examine how the imbalance of adaptation energy affects cross-task interference.

**Effect of smoothing on interference.** In the first experiment (Fig. 5, left), we evaluate performance on a target task by injecting LoRA adapters from the remaining tasks as perturbations. To modulate the degree of imbalance in these

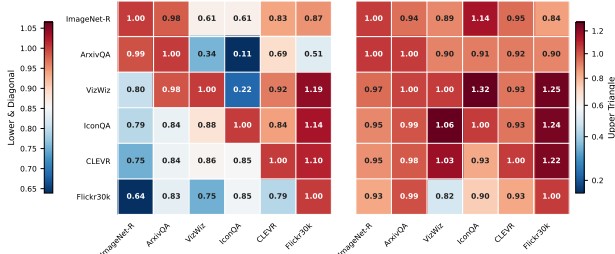

*Figure 6.* Heatmaps of LoRA (left) and EBO (right) in UCIT tasks. The upper triangle denotes the performance on unseen tasks, while the lower triangle denotes performance on seen tasks. Raw data for these visualizations can be found in Tab. 9a and 7.

perturbations, we apply a partial smoothing transformation to the distribution of their singular values: for a smoothing ratio $\alpha \in [0, 1]$, each singular value $\sigma_i$ is replaced by $(1 - \alpha)\sigma_i + \alpha\bar{\sigma}$, where $\bar{\sigma}$ is the mean singular value. We observe that increasing the smoothing ratio consistently improves performance across all target tasks, indicating that adapters with balanced singular value spectra exhibit lower interference when merged with existing knowledge. This provides empirical evidence that the highly skewed energy distribution observed in standard LoRA artifacts directly contributes to disruptive cross-task interactions.

**Effect of balanced updates on robustness.** In the second experiment (Fig. 5, right), we compare standard LoRA with the proposed EBO under equal-norm perturbations. For a given target task, we incrementally add LoRA adapters from other tasks, scaling their perturbation norm to match the

target update norm. While both methods suffer degradation under increasing interference, the decline for standard LoRA is substantially steeper. In contrast, EBO remains markedly more stable, suggesting that spectral balance increases the robustness of acquired knowledge to future interference.

**Interpretation.** Taken together, these results validate two complementary claims: (i) updates with balanced distribution of adaptation energy produce reduced cross-task interference onto previously acquired knowledge, and (ii) energy-balanced updates are themselves more robust to interference from subsequent tasks. These phenomena align with our component-based perspective, where catastrophic forgetting arises from uncontrolled competition among dominant components. By regulating energy-balance among components, the process of continual learning becomes more stable without sacrificing adaptability.

### 4.3. Ablation Study

In Fig. 6, we compare LoRA (Hu et al., 2022) and EBO variant introduced in Sec. 4.2 on UCIT. The diagonal reports the accuracy of each task immediately after it is learned; the lower triangle reflects the performance of previously learned tasks during the continual learning process (normalized w.r.t. the diagonal); and the upper triangle reflects generalization to unseen tasks during training (normalized w.r.t. zero-shot performance). This experiment is designed to isolate the effect of balanced energy on continual transfer by removing gradient-orthogonality and initialization components. As shown in Fig. 6, EBO produces noticeably higher values in the upper triangle and more stable values in the lower triangle compared to LoRA-FT, indicating reduced backward forgetting and improved forward transfer, respectively.

In Tab. 3, we further evaluate the contribution of individual components. EBO alone outperforms the other continual learning methods across all four aggregated metrics, demonstrating that energy-balanced updates substantially improve continual learning performance. Incorporating gradient orthogonality (GO) further reduces backward forgetting by steering updates away from previously sensitive modes. Finally, adding linear initialization (IL) improves forward transfer while preserving most of the stability gains, yielding the best overall performance.

**Free-spectrum variant.** To further examine whether the improvement comes from balanced singular value spectra rather than merely from using an orthogonal low-rank parameterization, we compare EBO with a free-spectrum variant, denoted as $\mathbf{U_t S_t V_t^\top}$, on the UCIT benchmark. This variant keeps the same orthonormal bases $\mathbf{U_t}$ and $\mathbf{V_t}$ as EBO, but replaces the scalar update magnitude $s_t$ with a learnable diagonal matrix $\mathbf{S_t}$, allowing different components to receive different singular values. Therefore, its spectrum

*Table 3.* Ablation results on UCIT. EBO denotes Energy-Balanced Optimization, GO denotes gradient orthogonality, and IL denotes the depth-aware initialization for $s_t$. The configuration without all three components corresponds to the LoRA-FT baseline.

| EBO | GO | IL | MFN↑ | MAA↑ | BWT↑ | FWT↑ |
|-----|-----|-----|------|------|------|------|
| ✗ | ✗ | ✗ | 61.4 | 76.5 | -15.4 | 26.8 |
| ✓ | ✗ | ✗ | 70.2 | 82.2 | -5.1 | 34.3 |
| ✓ | ✓ | ✗ | 72.1 | 82.5 | -2.6 | 34.0 |
| ✓ | ✓ | ✓ | **72.8** | **82.9** | **-2.0** | **34.6** |

*Table 4.* Comparison between EBO and the free-spectrum variant on the UCIT benchmark.

| Parameterization | MFN↑ | MAA↑ | BWT↑ | FWT↑ |
|-----|------|------|------|------|
| $\mathbf{U_t S_t V_t^\top}$ | 68.1 | 80.5 | -7.1 | 30.6 |
| $s_t \mathbf{U_t V_t^\top}$ (EBO) | **70.2** | **82.2** | **-5.1** | **34.3** |

can become imbalanced during optimization.

As shown in Tab. 4, the free-spectrum variant still clearly outperforms standard LoRA-FT in Tab. 3, suggesting that the orthogonal low-rank parameterization is itself beneficial. However, it underperforms EBO across all four aggregated metrics, indicating that explicitly enforcing a balanced singular value spectrum provides consistent gains beyond the orthogonal parameterization.

## 5. Related Works

Low-rank adaptation (LoRA) introduces trainable low-rank matrices to achieve parameter-efficient fine-tuning (Hu et al., 2022). Subsequent works further enhance its expressivity and optimization strategies. DoRA (Liu et al., 2024) decouples the magnitude and direction of updates. StelLA (Li et al., 2025b) employs Riemannian optimization to balance the update speeds of the $\mathbf{U}$ and $\mathbf{V}$ subspaces. However, such variants overlook the need of models to acquire new knowledge while preserving prior abilities. To mitigate forgetting, existing continual learning methods can be roughly categorized into three types: replay-based, regularization-based and architecture-based (Guo et al., 2025b).

**Replay-based approach.** Replay-based continual learning revisits past data to approximate joint training, and has been widely explored in multimodal settings. Early works replay raw image–question–answer tuples to preserve the performance on proier tasks, such as VQACL (Zhang et al., 2023) and MSPT (Chen et al., 2024). Recent works focus on selecting which samples to retain for replay, including clustering-based selection (Lin et al., 2025; Zhang et al., 2025), OASIS (Lee et al., 2025) and Adapt-∞ (Maharana et al., 2025). Despite these advances, replay methods still

require storing historical information, which poses storage and privacy concerns in real-world deployments.

**Regularization-based approach.** Regularization-based approaches alleviate forgetting by imposing constraints during training to preserve important parameters or representations from previous tasks. DAS (Ke et al., 2023) introduces a domain-adaptive pretraining framework with soft-masking to regulate updates based on unit importance. ARPER (Mi et al., 2020) builds on elastic weight consolidation by dynamically penalizing parameters according to vocabulary shifts, focusing protection on knowledge-critical weights across tasks. SEFE (Chen et al., 2025) distinguishes superficial from essential forgetting and applies parameter-level regularization to selectively stabilize critical LoRA updates during continual instruction tuning. In the context of full fine-tuning, Gradient Projection Memory (GPM) (Saha et al., 2021) enforces orthogonality between new-task gradients and a stored basis of principal gradient directions from previous tasks. Orthogonality has also been adopted in parameter-efficient tuning. O-LoRA (Wang et al., 2023a) constrains the LoRA subspaces of new tasks to be orthogonal to those of previous tasks. InfLoRA (Liang & Li, 2024) and KeepLoRA (Luo et al., 2026) constrain the LoRA down-projection matrix $\mathbf{A}$ to be orthogonal to GPM (Saha et al., 2021) or DualGPM (Liang & Li, 2023) to prevent interference.

**Architecture-based approach.** Architecture-based approaches freeze the pre-trained model and extend it with new parameters for each task. L2P (Wang et al., 2022b) selects the most relevant prompts from a prompt pool, while Dual-Prompt (Wang et al., 2022a) uses explicit task-sharing and task-specific prompts. CODA-Prompt (Smith et al., 2023) proposes end-to-end prompt selection methods to increase plasticity. MoE-Adapters (Yu et al., 2024) inserts a mixture of adapters into the image encoder, activating a subset for each task. DIKI (Tang et al., 2024) calibrates knowledge integration by determining the likelihood that a test sample belongs to a learned task. IAP (Fu et al., 2025) introduces Instance-Aware Gated Prompting to further improve prompt selection. CL-MoE (Huai et al., 2025) employs a dual-router mixture-of-experts and momentum-based expert fusion to preserve task-specific knowledge during continual learning. HiDe-LLaVA (Guo et al., 2025a) hierarchically decomposes LoRA modules into general and task-specific components to reduce forgetting. However, these methods cannot entirely avoid parameter selection errors or suboptimal activation coefficients.(Gu et al., 2026)

## 6. Conclusion

This work is motivated by the observation that low-rank updates exhibit highly imbalanced singular value spectra,

degrading previously acquired knowledge and making the updates more vulnerable to interference from subsequent tasks. Building on this insight, we proposed EBLoRA, which balances the distribution of adaptation energy in low-rank updates by decoupling the magnitude from directional structure, and constrains update directions to avoid interfering with previously acquired knowledge. Experiments show that EBLoRA significantly mitigates backward forgetting and improves forward transfer in continual learning settings while retaining a plasticity comparable to unconstrained LoRA on individual tasks. As a simple and effective method, EBLoRA provides a principled approach to low-rank continual adaptation that is compatible with standard deep-learning optimizers and scalable to larger models and diverse tasks.

## Acknowledgements

This work was supported by the National Science Foundation of China (62576092, 62225602), the Basic Research Program of Jiangsu (BK20253021), and the Big Data Computing Center of Southeast University. We would also like to sincerely thank the anonymous reviewers for their constructive suggestions.

## Impact Statement

This paper presents work whose goal is to advance the field of machine learning. There are many potential societal consequences of our work, none of which we feel must be specifically highlighted here.

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

# A. Proofs of Propositions

## A.1. Proofs of Proposition 3.1

*Proof.* Since $\mathcal{T}_{\mathbf{U}}\mathcal{M}_t$ is a linear subspace, the minimizer of $\min_{\mathbf{Y}\in\mathcal{T}_{\mathbf{U}}\mathcal{M}_t}\|\mathbf{Y}-\mathbf{Z}\|_F^2$ is unique and is characterized by the orthogonality condition: $\mathbf{Y}^\star$ is the orthogonal projection of $\mathbf{Z}$ onto $\mathcal{T}_{\mathbf{U}}\mathcal{M}_t$ if and only if $\mathbf{Z}-\mathbf{Y}^\star$ is orthogonal to $\mathcal{T}_{\mathbf{U}}\mathcal{M}_t$ under $\langle\mathbf{A},\mathbf{B}\rangle = \operatorname{tr}(\mathbf{A}^\top\mathbf{B})$.

We first verify feasibility. Let $\mathbf{Y}^\star = \mathbf{Z}_0 - \mathbf{U}\operatorname{sym}(\mathbf{U}^\top\mathbf{Z}_0)$. By construction, $\mathbf{G}^\top\mathbf{Z}_0 = \mathbf{0}$. Moreover, since $\mathbf{G}^\top\mathbf{U} = \mathbf{0}$ (because $\mathbf{U}\in\mathcal{M}_t$), we have $\mathbf{G}^\top\mathbf{Y}^\star = \mathbf{0}$. Next,

$$\mathbf{U}^\top\mathbf{Y}^\star + \mathbf{Y}^{\star\top}\mathbf{U} = \mathbf{U}^\top\mathbf{Z}_0 + \mathbf{Z}_0^\top\mathbf{U} - 2\operatorname{sym}(\mathbf{U}^\top\mathbf{Z}_0) = \mathbf{0},$$

so $\mathbf{Y}^\star\in\mathcal{T}_{\mathbf{U}}\mathcal{M}_t$.

It remains to prove optimality via orthogonality. Let the residual be $\mathbf{R} = \mathbf{Z}-\mathbf{Y}^\star$. Using $\mathbf{Z}_0 = (\mathbf{I}-\mathbf{G}\mathbf{G}^\top)\mathbf{Z}$, we can write

$$\mathbf{R} = \mathbf{Z}-\mathbf{Z}_0 + \mathbf{U}\operatorname{sym}(\mathbf{U}^\top\mathbf{Z}_0) = \mathbf{G}\mathbf{G}^\top\mathbf{Z} + \mathbf{U}\,\mathbf{S}, \quad \mathbf{S} = \operatorname{sym}(\mathbf{U}^\top\mathbf{Z}_0),$$

where $\mathbf{S}$ is symmetric.

Take any $\mathbf{Y}\in\mathcal{T}_{\mathbf{U}}\mathcal{M}_t$, i.e., $\mathbf{G}^\top\mathbf{Y} = \mathbf{0}$ and $\operatorname{sym}(\mathbf{U}^\top\mathbf{Y}) = \mathbf{0}$. We show $\langle\mathbf{R},\mathbf{Y}\rangle = 0$ by splitting the residual:

$$\langle\mathbf{G}\mathbf{G}^\top\mathbf{Z},\mathbf{Y}\rangle = \operatorname{tr}(\mathbf{Z}^\top\mathbf{G}\mathbf{G}^\top\mathbf{Y}) = \operatorname{tr}(\mathbf{Z}^\top\mathbf{G}(\mathbf{G}^\top\mathbf{Y})) = 0,$$

and we have:

$$\langle\mathbf{U}\mathbf{S},\mathbf{Y}\rangle = \operatorname{tr}(\mathbf{S}^\top\mathbf{U}^\top\mathbf{Y}) = \operatorname{tr}(\mathbf{S}\,\mathbf{U}^\top\mathbf{Y}).$$

Decompose $\mathbf{U}^\top\mathbf{Y}$ into symmetric and skew-symmetric parts: $\mathbf{U}^\top\mathbf{Y} = \operatorname{sym}(\mathbf{U}^\top\mathbf{Y}) + \operatorname{skew}(\mathbf{U}^\top\mathbf{Y})$. Since $\mathbf{S}$ is symmetric, we claim for any skew-symmetric matrices $B$ that $\operatorname{tr}(\mathbf{S}B) = 0$ through following process:

$$\operatorname{tr}(\mathbf{S}B) = \operatorname{tr}(B^\top\mathbf{S}^\top) = -\operatorname{tr}(B\mathbf{S}) = -\operatorname{tr}(\mathbf{S}B)$$

Combine with fact that $\operatorname{sym}(\mathbf{U}^\top\mathbf{Y}) = 0$ for all $\mathbf{Y}\in\mathcal{T}_{\mathbf{U}}\mathcal{M}_t$, we obtain $\langle\mathbf{U}\mathbf{S},\mathbf{Y}\rangle = 0$. Therefore $\langle\mathbf{R},\mathbf{Y}\rangle = 0$ for all $\mathbf{Y}\in\mathcal{T}_{\mathbf{U}}\mathcal{M}_t$, i.e., $\mathbf{R}\perp\mathcal{T}_{\mathbf{U}}\mathcal{M}_t$.

Hence, $\mathbf{Y}^\star$ is the orthogonal projection of $\mathbf{Z}$ onto $\mathcal{T}_{\mathbf{U}}\mathcal{M}_t$, and it is the unique minimizer of the stated problem. $\square$

## A.2. Proofs of Proposition 3.2

*Proof.* We first verify feasibility. By construction,

$$\mathbf{Y} = (\mathbf{I}-\mathbf{G}\mathbf{G}^\top)\widetilde{\mathbf{U}}$$

lies in the null space of $\mathbf{G}^\top$, since

$$\mathbf{G}^\top\mathbf{Y} = \mathbf{G}^\top(\mathbf{I}-\mathbf{G}\mathbf{G}^\top)\widetilde{\mathbf{U}} = (\mathbf{G}^\top - \mathbf{G}^\top\mathbf{G}\mathbf{G}^\top)\widetilde{\mathbf{U}} = \mathbf{0},$$

where we used $\mathbf{G}^\top\mathbf{G} = \mathbf{I}$. The retracted point is defined as

$$\mathbf{U}^+ = \mathbf{Y}(\mathbf{Y}^\top\mathbf{Y})^{-\frac{1}{2}},$$

which satisfies $\mathbf{U}^{+\top}\mathbf{U}^+ = \mathbf{I}_r$ by symmetry and positive-definiteness of $\mathbf{Y}^\top\mathbf{Y}$. Moreover, since right multiplication by an invertible $r\times r$ matrix preserves column spans, $\operatorname{Col}(\mathbf{U}^+) = \operatorname{Col}(\mathbf{Y}) \subseteq \operatorname{Null}(\mathbf{G}^\top)$, hence $\mathbf{G}^\top\mathbf{U}^+ = \mathbf{0}$. Therefore $\mathbf{U}^+\in\mathcal{M}_t$.

We now establish optimality. For any $\mathbf{U}\in\mathcal{M}_t$ we have $\mathbf{P}_{\perp\mathbf{G}}\mathbf{U} = \mathbf{U}$, where $\mathbf{P}_{\perp\mathbf{G}} = \mathbf{I}-\mathbf{G}\mathbf{G}^\top$ is the orthogonal projector onto $\operatorname{Null}(\mathbf{G}^\top)$, and

$$\widetilde{\mathbf{U}} = \mathbf{P}_{\perp\mathbf{G}}\widetilde{\mathbf{U}} + \mathbf{G}\mathbf{G}^\top\widetilde{\mathbf{U}} = \mathbf{Y} + \mathbf{G}\mathbf{G}^\top\widetilde{\mathbf{U}}.$$

Since $\mathbf{U}-\mathbf{Y}\in\operatorname{Null}(\mathbf{G}^\top)$ and $\mathbf{G}\mathbf{G}^\top\widetilde{\mathbf{U}}\in\operatorname{Col}(\mathbf{G})$ are orthogonal subspaces, we obtain the Pythagorean identity

$$\|\mathbf{U}-\widetilde{\mathbf{U}}\|_F^2 = \|\mathbf{U}-\mathbf{Y}\|_F^2 + \|\mathbf{G}\mathbf{G}^\top\widetilde{\mathbf{U}}\|_F^2.$$

The second term does not depend on $\mathbf{U}$, hence

$$\arg\min_{\mathbf{U}\in\mathcal{M}_t} \|\mathbf{U}-\widetilde{\mathbf{U}}\|_F^2 = \arg\min_{\mathbf{U}\in\mathcal{M}_t} \|\mathbf{U}-\mathbf{Y}\|_F^2.$$

We now minimize $\|\mathbf{U}-\mathbf{Y}\|_F^2$ over $\mathbf{U}$ satisfying only the orthogonality constraint $\mathbf{U}^\top\mathbf{U} = \mathbf{I}_r$. Expanding

$$\|\mathbf{U}-\mathbf{Y}\|_F^2 = \|\mathbf{U}\|_F^2 + \|\mathbf{Y}\|_F^2 - 2\,\mathrm{tr}(\mathbf{U}^\top\mathbf{Y}),$$

and noting $\|\mathbf{U}\|_F^2 = r$ is constant on the Stiefel manifold, the problem is equivalent to maximizing $\mathrm{tr}(\mathbf{U}^\top\mathbf{Y})$. Let the thin SVD of $\mathbf{Y}$ be

$$\mathbf{Y} = \mathbf{Q}\mathbf{\Sigma}\mathbf{P}^\top,$$

where $\mathbf{Q}\in\mathbb{R}^{d\times r}$ and $\mathbf{P}\in\mathbb{R}^{r\times r}$ are orthogonal, and $\mathbf{\Sigma} = \mathrm{diag}(\sigma_1,\ldots,\sigma_r)$ with $\sigma_i > 0$. Then

$$\mathrm{tr}(\mathbf{U}^\top\mathbf{Y}) = \mathrm{tr}(\mathbf{\Sigma}\,\mathbf{Z}), \quad \mathbf{Z} = \mathbf{Q}^\top\mathbf{U}\mathbf{P}.$$

Since $\mathbf{Z}$ is orthogonal whenever $\mathbf{U}$ is, von Neumann's trace inequality implies

$$\mathrm{tr}(\mathbf{\Sigma}\,\mathbf{Z}) \le \sum_{i=1}^{r} \sigma_i,$$

with equality if and only if $\mathbf{Z} = \mathbf{I}_r$. The maximizing point is therefore $\mathbf{U}^* = \mathbf{Q}\mathbf{P}^\top$, and since

$$\mathbf{Y}(\mathbf{Y}^\top\mathbf{Y})^{-\frac{1}{2}} = \mathbf{Q}\mathbf{\Sigma}\mathbf{P}^\top(\mathbf{P}\mathbf{\Sigma}^2\mathbf{P}^\top)^{-\frac{1}{2}} = \mathbf{Q}\mathbf{P}^\top,$$

we have $\mathbf{U}^+ = \mathbf{U}^*$.

Finally, since $\mathbf{U}^+$ also satisfies $\mathbf{G}^\top\mathbf{U}^+ = 0$, it is feasible for $\mathcal{M}_t$, and no smaller objective value can be attained over the subset $\mathcal{M}_t \subseteq \mathrm{St}(d,r)$. Uniqueness follows from strict positivity of $\mathbf{\Sigma}$ and the equality condition of von Neumann's inequality. Therefore $\mathbf{U}^+$ is the unique minimizer of the stated problem. $\square$

# B. Experiment Details

We adopt the LLaVA-1.5-7B (Liu et al., 2023) model for multimodal continual instruction tuning experiments. Training is performed on $4\times$ NVIDIA RTX PRO6000 Blackwell GPUs. Appendix B.1 provides further details of experiment settings and hyperparameters. Appendix B.2 introduces the datasets used in our experiments. Appendix B.3 describes the evaluation metrics for continual learning together with their definitions and computation formulas.

## B.1. Implementation Details

**Experiments on continual learning benchmarks.**   We integrate our projected optimization into AdamW by applying a pre-step gradient projection and a post-step update projection, followed by a retraction. For the MLLM-DCL benchmark, we set the learning rate to $2\times 10^{-5}$ and train for up to 5 epochs per task. For the UCIT benchmark, the learning rate is set to $2\times 10^{-4}$ for all tasks except Flickr30k, which uses $1\times 10^{-4}$ and train for up to 3 epochs per task.

We present the details of the depth-aware initialization for $s_t$ mentioned in Sec. 3.2 here. For each layer $\ell \in \{1,\ldots,L\}$, we initialize $s_t^{(\ell)}$ as following:

$$s_t^{(\ell,0)} = s_{\min}^{(0)} + \frac{\ell-1}{L-1}\left(s_{\max}^{(0)} - s_{\min}^{(0)}\right), \tag{17}$$

where $s_{\min}^{(0)}$ and $s_{\max}^{(0)}$ specify the initial scales for the shallowest and deepest layers, respectively. In our experiments, we set $s_{\min}^{(0)} = 0.002$ and $s_{\max}^{(0)} = 0.010$. This depth-aware initialization strategy stabilizes early training and improves convergence in practice.

**Experiments on model merging tasks.**   For the UCIT and MM-MergeBench settings, we follow the settings of direct merge. We fine-tune a separate LoRA adapter for each task individually for up to 3 epochs per task and sequentially add the LoRA weights from each adapter to the base model, yielding a model that aggregates task-specific low-rank updates. For the multimodal projector (mm_projector), we compute the element-wise mean across the participating adapters and replace the projector parameters with this averaged value. This produces a multi-task model capable of handling all merged tasks.

---

**Algorithm 2** Structured Continual Learning with Restricted-Manifold Updates

---

**Require:** Pre-trained weights $W_0$; stream of $T$ tasks rank $r$; optimizer OPT.

1: Initialize previous task gradient subspace $\mathbf{G}_0 \leftarrow \mathbf{0}$.
2: **for** $t = 1$ to $T$ **do**
3:   **// Gradient snapshot for task $\mathcal{T}_t$**
4:   Sample a small subset of mini-batches from task $t$ and form a gradient estimation $\mathcal{G}_t$ for initialization
5:   **// Projected low-rank initialization of $\Delta W_t$**
6:   $\mathbf{P}_{\perp \mathbf{G}_{t-1}} \leftarrow \mathbf{I} - \mathbf{G}_{t-1}\mathbf{G}_{t-1}^\top$
7:   $\mathcal{G}_t^{\text{proj}} \leftarrow \mathbf{P}_{\perp \mathbf{G}_{t-1}} \mathcal{G}_t$
8:   Compute SVD $\mathcal{G}_t^{\text{proj}} = U\Sigma V^\top$ and initialize $U_t^{(0)} \leftarrow U_{:,1:r}$, $V_t^{(0)} \leftarrow V_{:,1:r}$.
9:   Initialize $s_t^{(0)}$ with linear depth-dependent initialization.
10:   **// Optimization on the restricted manifold**
11:   Take $\mathbf{G}_{t-1}$ as the constraint and use Algorithm 1 to update $\mathbf{U}_t$ (with analogous projected updates for $\mathbf{V}_t$ and Euclidean updates for $s_t$) to minimize $\mathcal{L}_t\big(\mathbf{W}_{t-1} + s_t \mathbf{U}_t \mathbf{V}_t^\top\big)$ starting from $(s_t^{(0)}, \mathbf{U}_t^{(0)}, \mathbf{V}_t^{(0)})$.
12:   **// Apply the learned low-rank update**
13:   $\mathbf{W}_t \leftarrow \mathbf{W}_{t-1} + s_t \mathbf{U}_t \mathbf{V}_t^\top$.
14:   **// Update previous-task knowledge subspace**
15:   Use GPM to update $\mathbf{G}_{t-1}$ with $\mathcal{G}_t$ to obtain $\mathbf{G}_t$
16: **end for**

---

## B.2. Benchmark

**MLLM-DCL** benchmark (Zhao et al., 2025) consists of multiple downstream VQA datasets: RSVQA (Lobry et al., 2020), PathVQA (He et al., 2020), DriveLM (Sima et al., 2024), FinVis (Wang et al., 2023b), AI2D (Kembhavi et al., 2016), SciVerse (Guo et al., 2025d), MapQA (Chang et al., 2022), and TQA (Kembhavi et al., 2017). It covers 5 specialized areas: Remote Sensing, Medical, Driving, Finance, and Science. Each area is treated as a task.

**UCIT** benchmark (Guo et al., 2025a) consists of 6 VQA datasets: ArxivQA (Li et al., 2024), CLEVR-Math (Lindström & Abraham, 2022), IconQA (Lu et al., 2021), ImageNet-R (Hendrycks et al., 2021), VizWiz-Caption (Gurari et al., 2018), and Flickr30k (Plummer et al., 2015). Each dataset is treated as a task.

**MM-MergeBench** (Zeng et al., 2025) benchmark contains 8 multimodal datasets as seen tasks: ScienceQA (Lu et al., 2022), ImageNet (Deng et al., 2009), VQAv2 (Goyal et al., 2017), REC-COCO (Kazemzadeh et al., 2014; Mao et al., 2016), OCRVQA (Mishra et al., 2019), Flickr30k (Plummer et al., 2015), VizWiz-caption (Gurari et al., 2018) and IconQA (Lu et al., 2021). Each dataset is treated as a task.

## B.3. Evaluation Metrics

We present the definitions and mathematical expressions of the four aggregate metrics employed in our evaluation: Mean Final Accuracy (MFN), Mean Average Accuracy (MAA), Backward Transfer (BWT), and Forward Transfer (FWT).

MFN measures the average accuracy across all tasks after the model has completed the entire incremental training sequence. This metric reflects the model's final performance after all tasks have been learned. It is defined as:

$$\text{MFN} = \frac{1}{T}\sum_{i=1}^{T} A_{T,i}, \tag{18}$$

where $A_{T,i}$ represents the accuracy on task $i$ after learning all $T$ tasks.

MAA provides a comprehensive measure of the model's performance throughout the entire training process. It is the mean of the average accuracies on all learned tasks after each incremental training step. The formula for MAA is:

$$\text{MAA} = \frac{1}{T}\sum_{j=1}^{T}\left(\frac{1}{j}\sum_{i=1}^{j} A_{j,i}\right), \tag{19}$$

where $A_{j,i}$ denotes the accuracy on task $i$ after the model has been trained on the first $j$ tasks.

*Table 5.* Training cost comparison between LoRA and EBLoRA on UCIT and MLLM-DCL. Training time is measured in seconds per iteration. Memory usage denotes peak GPU memory.

| Metric | Method | UCIT | MLLM-DCL |
|---|---|---|---|
| Training time (s/it) | LoRA | 3.28 | 3.62 |
| Training time (s/it) | EBLoRA | 4.20 | 4.51 |
| Training time ratio | EBLoRA / LoRA | 1.28× | 1.25× |
| Memory usage (GB) | LoRA | 32.31 | 35.52 |
| Memory usage (GB) | EBLoRA | 35.53 | 37.42 |
| Memory usage ratio | EBLoRA / LoRA | 1.10× | 1.05× |
| Mean aggregated score (%) | LoRA | 37.33 | 34.40 |
| Mean aggregated score (%) | EBLoRA | 47.07 | 42.15 |
| Mean aggregated score ratio | EBLoRA / LoRA | 1.26× | 1.23× |

BWT assesses the extent of forgetting by measuring the difference in accuracy for each task between the final training step and immediately after the task was learned. It is calculated as:

$$\text{BWT} = \frac{1}{T} \sum_{i=1}^{T} \left( A_{T,i} - A_{i,i} \right), \tag{20}$$

A negative BWT value indicates forgetting, with larger negative values implying greater forgetting.

FWT quantifies the model's ability to generalize learned knowledge to future unseen tasks. Formally, it is defined as:

$$\text{FWT} = \frac{1}{T-1} \sum_{i=2}^{T} \left( \frac{1}{i-1} \sum_{j=1}^{i-1} A_{j,i} \right), \tag{21}$$

where $A_{j,i}$ denotes the accuracy on task $i$ after the model has been trained on the first $j$ tasks.

Additionally, we present Average Accuracy (AVG) for the evaluation in Appendix D. AVG is the overall mean accuracy of both seen and unseen tasks through all training steps. It reflects the overall performance and the stability of the training process. AVG is defined as:

$$\text{AVG} = \frac{1}{T^2} \sum_{j=1}^{T} \sum_{i=1}^{T} A_{j,i}. \tag{22}$$

### B.4. Training Cost

We further report the computational and memory overhead of EBLoRA compared with standard LoRA. Training time is measured as the average wall-clock time per iteration, and memory usage is measured as the peak GPU memory during training. To summarize the overall performance gain in a single scalar, we report the mean aggregate score, computed as the arithmetic mean of MFN, MAA, BWT, and FWT. Since all four metrics are higher-is-better, a larger mean aggregated score indicates better overall continual learning performance.

As shown in Tab. 5, EBLoRA introduces moderate overhead compared with LoRA, requiring about $1.25\times$ training time and $1.05$–$1.10\times$ GPU memory. Meanwhile, it yields substantially larger continual learning gains, achieving about $1.23$–$1.26\times$ improvement in the mean aggregate score. These results indicate that the additional cost of projected optimization and retraction is moderate relative to the performance improvement.

**Newton–Schulz orthogonalization.** The whitening-based retraction used in Eq. 12 maps a tentative update back to the restricted Stiefel manifold by

$$\mathbf{Y} = \mathbf{P}_{\perp \mathbf{G}} \widetilde{\mathbf{U}}, \qquad \mathbf{U}^+ = \mathbf{Y}(\mathbf{Y}^\top \mathbf{Y})^{-\frac{1}{2}}, \tag{23}$$

where $\mathbf{P}_{\perp \mathbf{G}} = \mathbf{I} - \mathbf{G}\mathbf{G}^\top$. Although this operation is performed on a small $r \times r$ matrix, computing the inverse square root still introduces additional overhead. We therefore evaluate an alternative implementation based on Newton–Schulz

*Table 6.* Accuracy of EBLoRA with Newton–Schulz orthogonalization on the UCIT benchmark. Each row represents the performance on every dataset of the model trained after the corresponding task. FWT , AVG , and MFN metrics are shown.

|  | ImgNet-R | ArxivQA | VizWiz | IconQA | CLEVR | Flickr30k |
|---|---|---|---|---|---|---|
| Transfer |  | 48.7 | 33.9 | 25.1 | 19.1 | 44.8 | 34.3 |
| ImgNet-R | 90.5 | 48.7 | 33.9 | 22.1 | 19.4 | 33.7 |
| ArxivQA | 90.6 | 93.5 | 33.9 | 25.1 | 19.7 | 34.1 |
| VizWiz | 89.9 | 94.1 | 61.5 | 28.2 | 19.2 | 52.3 |
| IconQA | 90.0 | 94.1 | 61.9 | 82.1 | 18.3 | 52.1 |
| CLEVR | 89.9 | 93.6 | 61.2 | 77.1 | 69.4 | 52.1 |
| Flickr30k | 88.4 | 94.0 | 47.4 | 77.5 | 66.2 | 56.3 | 71.6 |
| Average | 89.9 | 86.3 | 50.0 | 52.0 | 35.4 | 46.7 | 60.1 |

orthogonalization (Schulz, 1933; Higham, 1986), which approximates the polar factor using matrix multiplications instead of explicit eigendecomposition.

Given the projected matrix $\mathbf{Y} = \mathbf{P}_{\perp \mathbf{G}}\tilde{\mathbf{U}}$, we initialize

$$\mathbf{X}_0 = \frac{\mathbf{Y}}{\|\mathbf{Y}\|_F}, \tag{24}$$

and iteratively apply

$$\mathbf{X}_{k+1} = \frac{1}{2}\mathbf{X}_k \left(3\mathbf{I}_r - \mathbf{X}_k^\top \mathbf{X}_k\right). \tag{25}$$

This iteration pushes the singular values of $\mathbf{X}_k$ toward one, so that $\mathbf{X}_k^\top \mathbf{X}_k \approx \mathbf{I}_r$ after a small number of iterations. Moreover, since every iterate $\mathbf{X}_k$ is obtained from $\mathbf{Y}$ through right multiplication by an $r \times r$ matrix, its column space remains within $\mathrm{Col}(\mathbf{Y}) \subseteq \mathrm{Null}(\mathbf{G}^\top)$ under exact arithmetic. Thus, Newton–Schulz orthogonalization approximately enforces the Stiefel constraint while preserving the gradient-orthogonality constraint.

Compared with the original whitening-based implementation in Tab. 9f, Newton–Schulz orthogonalization yields slightly lower final performance overall, but leads to faster training in practice. Its training time per iteration is about $0.85\times$ that of the whitening-based EBLoRA implementation and about $1.05\times$ that of standard LoRA. This suggests that Newton–Schulz orthogonalization is a viable alternative when training efficiency is preferred.

## C. Details of Gradient Projection Memory

GPM (Saha et al., 2021) is a well-established method that maintains a subspace of task-sensitive gradient directions for mitigating interference. In our method, GPM is used to accumulate gradient information from past tasks. For clarity and completeness, we detail the procedure in this section.

**Initialization.** Since no prior tasks exist at $t = 1$, the subspace is initialized as $\mathbf{G}_0 = \mathbf{0}$.

**Gradient collection.** For task $t$, we compute an aggregated gradient snapshot $\mathcal{G}_t$ and project it onto the orthogonal complement of the stored subspace: $\mathcal{G}_t^\perp = (\mathbf{I} - \mathbf{G}_{t-1}\mathbf{G}_{t-1}^\top)\mathcal{G}_t$.

**Subspace update.** Let the SVD of $\mathcal{G}_t^\perp$ be $\mathcal{G}_t^\perp = \mathbf{U}\boldsymbol{\Sigma}\mathbf{V}^\top$ with singular values $\{\sigma_i\}$. Following (Saha et al., 2021), the smallest $r_t$ is retained such that the selected components capture an $\epsilon$-fraction of the gradient energy:

$$\left\|\mathbf{G}_{t-1}^\top \mathcal{G}_t\right\|_F^2 + \sum_{i=1}^{r_t} \sigma_i^2 \geq \epsilon\|\mathcal{G}_t\|_F^2,$$

where $\epsilon = 0.95$ is a threshold hyperparameter. The corresponding basis vectors are appended: $\mathbf{G}_t = \begin{bmatrix} \mathbf{G}_{t-1} & \mathbf{U}_{:,1:r_t} \end{bmatrix}$.

**Usage.** For subsequent tasks, the stored subspace imposes the constraint $\mathbf{G}_{t-1}^\top \mathbf{U}_t = \mathbf{0}$, which prevents new updates from interfering with previously important gradient modes.

## D. Additional Experiments

We present the detailed per-training-step accuracies through all training steps in Tab. 7, 8, 9 and 10. Results in Tab. 9 and 10 demonstrate strong performance of EBLoRA in terms of both learning plasticity and stability, while results in Tab. 7 and 8 confirms the efficacy of our proposed EBO, supporting our claim that maintaining balanced singular value spectra reduces cross-task interference during the learning process.

*Table 7.* Accuracy of EBO on the UCIT benchmark. Each row represents the performance on every dataset of the model trained after the corresponding task. FWT , AVG , and MFN metrics are shown.

|          | ImgNet-R | ArxivQA | VizWiz | IconQA | CLEVR | Flickr30k |
|----------|------|------|------|------|------|------|
| Transfer |      | 50.5 | 34.5 | 21.6 | 19.2 | 45.7 | 34.3 |
| ImgNet-R | 90.6 | 50.5 | 34.2 | 21.9 | 19.6 | 35.4 |
| ArxivQA  | 90.9 | 94.1 | 34.7 | 17.5 | 18.9 | 37.6 |
| VizWiz   | 88.0 | 93.7 | 61.0 | 25.3 | 19.2 | 52.3 |
| IconQA   | 86.4 | 92.8 | 64.9 | 82.8 | 19.2 | 51.9 |
| CLEVR    | 85.9 | 92.5 | 63.1 | 77.2 | 67.3 | 51.2 |
| Flickr30k | 84.6 | 93.0 | 49.7 | 74.9 | 62.8 | 55.9 | 70.2 |
| Average  | 87.7 | 86.1 | 51.2 | 49.9 | 34.5 | 47.4 | 59.5 |

*Table 8.* Accuracy of EBO on the MLLM-DCL benchmark. Each row represents the performance on every dataset of the model trained after the corresponding task. FWT , AVG , and MFN metrics are shown.

|          | Scesing | Medical | Driving | Science | Finance |
|----------|------|------|------|------|------|
| Transfer |      | 29.4 | 19.1 | 32.6 | 50.3 | 32.5 |
| Sensing  | 80.7 | 29.4 | 19.5 | 34.4 | 57.1 |
| Medical  | 73.6 | 62.0 | 18.7 | 31.6 | 50.9 |
| Driving  | 77.2 | 58.5 | 54.0 | 31.9 | 47.2 |
| Science  | 78.6 | 49.1 | 47.3 | 53.9 | 46.0 |
| Finance  | 77.2 | 50.9 | 41.7 | 51.0 | 89.5 | 62.1 |
| Average  | 77.5 | 50.0 | 36.2 | 40.6 | 58.1 | 52.5 |

**Use of Large Language Models**

We use the large language model to polish text and check grammar. All outputs were reviewed by the authors, who take full responsibility for the final content.

*Table 9.* Accuracy of LoRA-FT, O-LoRA, CL-MoE, SEFE, KeepLoRA and EBLoRA on the UCIT benchmark. Each row represents the performance on every dataset of the model trained after the corresponding task.  FWT , AVG , and MFN  metrics are shown.

### (a) LoRA-FT

|  | ImgNet-R | ArxivQA | VizWiz | IconQA | CLEVR | Flickr30k |
|---|---|---|---|---|---|---|
| Transfer |  | 52.6 | 18.3 | 6.0 | 17.0 | 40.3 26.8 |
| ImgNet-R | 91.7 | 52.6 | 23.5 | 11.8 | 17.2 | 36.5 |
| ArxivQA | 90.5 | 92.1 | 13.1 | 2.1 | 14.2 | 21.5 |
| VizWiz | 73.6 | 90.7 | 61.0 | 4.2 | 19.0 | 49.7 |
| IconQA | 72.7 | 77.1 | 53.7 | 79.7 | 17.4 | 47.8 |
| CLEVR | 68.8 | 77.4 | 52.3 | 67.8 | 77.9 | 46.1 |
| Flickr30k | 58.6 | 76.7 | 45.7 | 67.4 | 61.6 | 58.0 61.4 |
| Average | 76.0 | 77.8 | 41.6 | 38.8 | 34.6 | 43.3 52.0 |

### (b) O-LoRA

|  | ImgNet-R | ArxivQA | VizWiz | IconQA | CLEVR | Flickr30k |
|---|---|---|---|---|---|---|
| Transfer |  | 52.9 | 19.6 | 4.4 | 16.9 | 41.0 27.0 |
| ImgNet-R | 91.5 | 52.9 | 24.7 | 13.3 | 17.3 | 36.5 |
| ArxivQA | 89.7 | 94.2 | 14.5 | 0.0 | 12.9 | 25.0 |
| VizWiz | 80.9 | 91.7 | 59.8 | 0.0 | 19.6 | 49.0 |
| IconQA | 80.2 | 80.3 | 54.5 | 75.9 | 17.6 | 48.6 |
| CLEVR | 78.1 | 80.4 | 51.6 | 63.2 | 72.4 | 46.0 |
| Flickr30k | 74.2 | 80.9 | 45.3 | 62.9 | 63.8 | 57.2 64.1 |
| Average | 82.4 | 80.1 | 41.7 | 35.9 | 33.9 | 43.7 53.0 |

### (c) CL-MoE

|  | ImgNet-R | ArxivQA | VizWiz | IconQA | CLEVR | Flickr30k |
|---|---|---|---|---|---|---|
| Transfer |  | 52.0 | 19.3 | 7.4 | 17.8 | 41.3 27.6 |
| ImgNet-R | 91.2 | 52.0 | 23.9 | 5.2 | 15.6 | 36.9 |
| ArxivQA | 89.2 | 92.5 | 14.8 | 10.0 | 15.7 | 26.2 |
| VizWiz | 77.2 | 90.7 | 60.4 | 6.9 | 20.6 | 49.5 |
| IconQA | 79.5 | 76.2 | 51.0 | 54.7 | 19.4 | 47.9 |
| CLEVR | 76.7 | 75.4 | 48.1 | 52.6 | 73.0 | 45.9 |
| Flickr30k | 61.2 | 75.8 | 44.4 | 52.6 | 54.4 | 57.3 58.6 |
| Average | 80.2 | 77.1 | 40.4 | 30.3 | 33.1 | 44.0 50.9 |

### (d) SEFE

|  | ImgNet-R | ArxivQA | VizWiz | IconQA | CLEVR | Flickr30k |
|---|---|---|---|---|---|---|
| Transfer |  | 53.3 | 18.7 | 7.5 | 17.0 | 40.9 27.5 |
| ImgNet-R | 91.6 | 53.3 | 23.7 | 12.1 | 16.9 | 36.4 |
| ArxivQA | 90.4 | 92.8 | 13.7 | 5.0 | 16.4 | 21.1 |
| VizWiz | 83.6 | 89.3 | 61.4 | 5.3 | 18.6 | 49.8 |
| IconQA | 84.3 | 78.1 | 57.4 | 79.6 | 16.2 | 50.6 |
| CLEVR | 82.8 | 78.6 | 54.2 | 70.6 | 75.0 | 46.5 |
| Flickr30k | 80.2 | 79.1 | 47.1 | 69.4 | 65.7 | 57.3 66.5 |
| Average | 85.5 | 78.6 | 42.9 | 40.3 | 34.8 | 43.6 54.3 |

### (e) KeepLoRA

|  | ImgNet-R | ArxivQA | VizWiz | IconQA | CLEVR | Flickr30k |
|---|---|---|---|---|---|---|
| Transfer |  | 52.8 | 20.4 | 9.2 | 18.1 | 41.5 28.4 |
| ImgNet-R | 91.5 | 52.8 | 25.6 | 13.4 | 17.1 | 36.7 |
| ArxivQA | 90.4 | 94.5 | 15.2 | 4.0 | 17.2 | 21.5 |
| VizWiz | 85.5 | 92.4 | 61.5 | 10.1 | 21.0 | 50.6 |
| IconQA | 85.1 | 86.0 | 55.7 | 76.9 | 17.1 | 50.9 |
| CLEVR | 84.1 | 89.3 | 51.5 | 68.3 | 72.6 | 47.8 |
| Flickr30k | 82.4 | 86.7 | 46.6 | 67.8 | 66.4 | 57.2 67.8 |
| Average | 86.5 | 83.6 | 42.7 | 40.1 | 35.2 | 44.1 55.4 |

### (f) EBLoRA

|  | ImgNet-R | ArxivQA | VizWiz | IconQA | CLEVR | Flickr30k |
|---|---|---|---|---|---|---|
| Transfer |  | 49.2 | 34.5 | 24.3 | 19.0 | 45.9 34.6 |
| ImgNet-R | 90.5 | 49.2 | 34.6 | 21.6 | 19.8 | 36.8 |
| ArxivQA | 90.4 | 94.5 | 34.3 | 22.6 | 19.5 | 36.3 |
| VizWiz | 90.0 | 94.4 | 61.5 | 28.7 | 19.0 | 52.0 |
| IconQA | 89.9 | 94.1 | 63.4 | 80.4 | 17.8 | 52.2 |
| CLEVR | 89.7 | 94.2 | 62.8 | 75.1 | 66.9 | 52.3 |
| Flickr30k | 89.6 | 94.2 | 56.6 | 75.2 | 66.2 | 55.3 72.8 |
| Average | 90.0 | 86.8 | 52.2 | 50.6 | 34.9 | 47.5 60.3 |

*Table 10.* Accuracy of LoRA-FT, O-LoRA, CL-MoE, SEFE, KeepLoRA and EBLoRA on the MLLM-DCL benchmark. Each row represents the performance on every dataset of the model trained after the corresponding task. FWT , AVG , and MFN metrics are shown.

*(a) LoRA-FT*

|  | Sensing | Medical | Driving | Science | Finance |  |
|---|---|---|---|---|---|---|
| Transfer |  | 28.1 | 17.4 | 34.0 | 50.2 | 32.4 |
| Sensing | 78.8 | 28.1 | 17.3 | 34.8 | 55.6 |  |
| Medical | 75.5 | 58.4 | 17.5 | 32.7 | 54.8 |  |
| Driving | 70.0 | 47.5 | 52.3 | 34.6 | 40.9 |  |
| Science | 73.2 | 46.4 | 40.6 | 50.4 | 49.5 |  |
| Finance | 69.3 | 44.3 | 29.1 | 41.4 | 88.4 | 54.5 |
| Average | 73.3 | 44.9 | 31.4 | 38.8 | 57.8 | 49.3 |

*(b) O-LoRA*

|  | Sensing | Medical | Driving | Science | Finance |  |
|---|---|---|---|---|---|---|
| Transfer |  | 28.4 | 18.4 | 33.7 | 52.5 | 33.3 |
| Sensing | 79.4 | 28.4 | 17.6 | 34.9 | 56.1 |  |
| Medical | 74.3 | 58.5 | 19.2 | 33.2 | 56.0 |  |
| Driving | 74.7 | 48.3 | 52.6 | 33.1 | 45.2 |  |
| Science | 74.6 | 46.5 | 42.2 | 50.1 | 52.8 |  |
| Finance | 72.3 | 46.9 | 31.6 | 41.5 | 88.1 | 56.1 |
| Average | 75.0 | 45.7 | 32.6 | 38.5 | 59.6 | 50.3 |

*(c) CL-MoE*

|  | Sensing | Medical | Driving | Science | Finance |  |
|---|---|---|---|---|---|---|
| Transfer |  | 28.3 | 19.4 | 34.1 | 48.6 | 32.6 |
| Sensing | 79.4 | 28.3 | 18.7 | 35.2 | 56.4 |  |
| Medical | 74.8 | 60.7 | 20.1 | 32.4 | 54.9 |  |
| Driving | 74.0 | 44.3 | 52.1 | 34.7 | 39.6 |  |
| Science | 71.0 | 47.4 | 40.0 | 50.7 | 43.3 |  |
| Finance | 71.8 | 47.4 | 29.5 | 41.5 | 89.2 | 55.9 |
| Average | 74.2 | 45.6 | 32.1 | 38.9 | 56.7 | 49.5 |

*(d) SEFE*

|  | Sensing | Medical | Driving | Science | Finance |  |
|---|---|---|---|---|---|---|
| Transfer |  | 28.1 | 19.6 | 33.9 | 52.4 | 33.5 |
| Sensing | 78.8 | 28.1 | 18.6 | 35.1 | 56.2 |  |
| Medical | 77.1 | 59.5 | 20.7 | 33.0 | 55.7 |  |
| Driving | 77.8 | 51.6 | 52.5 | 33.5 | 47.4 |  |
| Science | 77.9 | 48.4 | 44.7 | 50.4 | 50.1 |  |
| Finance | 77.1 | 50.9 | 40.3 | 43.0 | 88.4 | 59.9 |
| Average | 77.7 | 47.7 | 35.4 | 39.0 | 59.6 | 51.9 |

*(e) KeepLoRA*

|  | Sensing | Medical | Driving | Science | Finance |  |
|---|---|---|---|---|---|---|
| Transfer |  | 28.5 | 16.6 | 34.1 | 55.6 | 33.7 |
| Sensing | 80.0 | 28.5 | 17.0 | 35.1 | 55.1 |  |
| Medical | 79.9 | 58.6 | 16.3 | 33.7 | 55.6 |  |
| Driving | 79.8 | 57.7 | 53.1 | 33.7 | 54.6 |  |
| Science | 79.2 | 54.9 | 51.1 | 51.6 | 57.2 |  |
| Finance | 78.8 | 54.3 | 50.2 | 49.5 | 89.3 | 64.4 |
| Average | 79.6 | 50.8 | 37.5 | 40.7 | 62.4 | 54.2 |

*(f) EBLoRA*

|  | Sensing | Medical | Driving | Science | Finance |  |
|---|---|---|---|---|---|---|
| Transfer |  | 29.2 | 19.3 | 34.3 | 54.6 | 34.4 |
| Sensing | 80.7 | 29.2 | 19.3 | 34.5 | 57.5 |  |
| Medical | 79.0 | 58.6 | 19.4 | 33.9 | 54.0 |  |
| Driving | 79.3 | 57.9 | 54.1 | 34.7 | 55.5 |  |
| Science | 78.4 | 58.1 | 54.0 | 53.0 | 51.5 |  |
| Finance | 79.4 | 57.2 | 53.9 | 52.5 | 90.6 | 66.7 |
| Average | 79.4 | 52.2 | 40.1 | 41.7 | 61.8 | 55.0 |

