# OpenReview forum: "Spectral Imbalance Causes Forgetting in Low-Rank Continual Adaptation"
_ICML.cc/2026/Conference — ICML 2026 regular_

### Official Review · Reviewer_EN9A · 2026-03-01

**Soundness:** 2
**Presentation:** 2
**Significance:** 3
**Originality:** 3
**Overall Recommendation:** 4
**Confidence:** 4

**Summary:**

This paper addresses the problem of continual learning in Vision-Language Models (VLMs) using low-rank adaptation (LoRA). The authors identify a spectral imbalance in LoRA weight updates—where a few singular directions carry disproportionately large energy—leading to forward and backward interference across tasks. To address this, they propose decoupling the magnitude and direction of updates and constraining the update directions to lie on the Stiefel manifold. A magnitude-direction factorization is introduced where a scalar controls "global adaptation energy" per layer, while directional factors are kept orthogonal to previous task subspaces via Riemannian optimization. The method uses standard optimizers with projection and retraction steps to maintain manifold constraints.

**Compliance With Llm Reviewing Policy:**

Affirmed.

**Final Justification:**

The paper has technical merits with weaknesses mainly related to presentation and clear explaination of the method. During rebuttal , Authors acknowledged the issues and provided list of changes they will make in revision. So I am raising my score.

**Key Questions For Authors:**

1. **Update-space vs. gradient-space projection:** What happens if orthogonal projection is applied only to the final update rather than for both gradient and update in Alg 1. An ablation isolating this choice would clarify the necessity of gradient-level projection.

2. **Task accuracy reporting:** Are the per-task accuracies reported after first encountering each task or after completing training on all tasks? What is the task ordering used? This is critical for interpreting the results and comparing with baselines.

3. **Flickr performance:** Can you provide analysis or intuition for why performance on Flickr is notably poor compared to other benchmarks?

4. **Newton shulz orthogonalization:** Can the whitening step in the eqn 12 use newton schulz orthogonalization ?

**Limitations:**

yes

**Strengths And Weaknesses:**

### Strengths

- **Originality:** The spectral analysis of LoRA updates and the identification of energy imbalance as a source of interference is a well-motivated and insightful contribution. Decoupling magnitude and direction on the Stiefel manifold is a principled approach to constraining continual low-rank adaptation.
- **Soundness:** The use of Riemannian optimization with projection and retraction to enforce manifold constraints is technically sound. The model merging analysis to characterize interference is a useful diagnostic.
- **Significance:** The problem of catastrophic forgetting in VLM fine-tuning is practically important. The proposed approach offers a geometrically grounded alternative to existing regularization-based continual learning methods.

### Weaknesses

- **Presentation:** The paper suffers from several presentation issues that hinder readability:
  - Key terms are introduced without precise definitions. For example, "global adaptation energy" is not clearly defined to show what it really means. The term $L_t$ in Equation 7 is left undefined.
  - "UCIT" is mentioned before being cited.
  - Figure 1 is difficult to parse; its caption does not explain the content or cite the compared methods.
  - The notation is inconsistent: Equation 9 uses $P^{\perp}_{G_{t-1}} = I - G_{t-1}G_{t-1}^\top$, but Equation 11 writes the full expression again instead of reusing the defined symbol.
  - The term "gradient orthogonality" is misleading. The constraint in Equation 9 enforces orthogonality of update directions to previous task subspaces, not orthogonality of the gradients themselves. This conflation is confusing, particularly in the ablation section.
  - The ablation introduces "Energy-Balanced Optimization (EBO)" without prior naming or clear definition, making it difficult to understand what is being isolated. The meaning of "energy-balancing mechanism" is unclear (line 302).
- **Soundness (concerns):**
  - The "gradient snapshot" mentioned at line 236 lacks explanation of how it is computed.
  - There is no experiment isolating the effect of projecting only the final update (rather than the gradient) onto the manifold in Algorithm 1.
- **Significance (limited evaluation):**
  - previous methods such as SPU [Zhang et al., CVPR 2024] and ZSCL [Zheng et al., ICCV 2023] are absent from the experimental comparisons.
  - The results discussion is thin. Poor performance on Flickr is not analyzed. The "accuracy on each task" metric is ambiguous—it is unclear whether this is measured after first encountering the task or after completing all training. The continual learning setup is unclear. The usage of replay buffer or size of the buffer is not mentioned. The task ordering is not specified.
- **Missing related work:** Sparsity-based methods (e.g., SPU) are not mentioned among regularization-based approaches in the related work section.

[1] SPU - Zhang, Wenxuan, et al. "Overcoming generic knowledge loss with selective parameter update." CVPR2024.

[2] ZSCL - Zheng et al Preventing zero-shot transfer degradation in continual learning of vision-language models ICCV 2023

---

> ### Author Rebuttal · Authors · 2026-03-29
>
> We thank the reviewer for the insightful feedback.
>
> ## W1: Presentation
> We thank the reviewer for pointing these out.
>
> We will define missing terms at first use, including **global adaptation energy** and $L_t$ in Eq. 7, cite UCIT before its first appearance, and revise Fig. 1 and its caption for clarity.
>
> We will make the notation more consistent, e.g., reuse the projector symbol once defined and replace **gradient orthogonality** with more precise wording.
>
> In Sec. 4.2, we will name EBO at first use, define it as the variant without Eq. 4 so as to isolate the energy-balancing component, and explicitly state that the “energy-balancing mechanism” refers to the $sUV^\top$ factorization that enforces a balanced spectrum.
> ## W2/Q1: Gradient snapshot and tangent-space projection
>
> The **gradient snapshot** refers to a small set of gradients collected on the current task to estimate its representative update subspace. In practice, we compute gradients on several mini-batches at the beginning of task $t$, flatten the per-layer gradients, and use them to form $G_t$, which is then projected onto the null space of the stored previous-task subspace and used to initialize the structured update basis. Appendix C contains related implementation details, and we will clarify this connection in the main text.
>
> Orthogonality is enforced at the gradient/tangent level because this is required by the restricted Stiefel geometry. By Proposition 3.1, our operator is the unique orthogonal projection onto the tangent space $T_U\mathcal{M}_t$ under the Frobenius norm, so the projected gradient in Algorithm 1 is the canonical Riemannian gradient. Projecting only the final update is only a post-hoc feasibility correction.
>
> |EBLoRA|MFN $\uparrow$|MAA $\uparrow$|BWT $\uparrow$|FWT $\uparrow$|
> |---|---:|---:|---:|---:|
> |w/ tangent-space ortho.|66.7|68.2|-0.7|34.4|
> |w/o tangent-space ortho.|65.3|66.9|-0.6|34.2|
>
> Removing tangent-space orthogonality lowers overall continual performance. Thus, the benefit is not merely obtaining a feasible final update, but ensuring that the optimization direction itself respects the restricted Stiefel geometry.
>
> ## W3/Q2: Reporting and order
>
> In the main tables, “accuracy on each task” refers to the **final accuracy after the full continual learning sequence**. The full training trajectory is provided in Appendix D, and the main-text tables follow the same setting and task ordering as the MCITLib benchmark (Guo et al., arXiv 2025). We will make this explicit in the revision. We also clarify that **no replay buffer is used**.
>
> ## W3/Q3: Flickr30k
>
> We thank the reviewer for this question. We agree that the relatively lower performance on Flickr30k is a limitation of the current method and should be discussed more clearly.
>
> In our setting, Flickr30k is the last task in the UCIT sequence, so its final performance is essentially determined by its own task fitting rather than additional forgetting from later tasks. This suggests that the issue is not mainly continual interference after learning Flickr30k, but that our method is somewhat less effective on this dataset itself.
>
> Our current understanding is that this is likely due to dataset-specific optimization difficulty rather than the spectrum-balancing design inherently limiting representation capacity. We will make this limitation explicit in the paper and add a more careful discussion of dataset-specific behavior.
>
> ## W3/W4: Missing related work
>
> We thank the reviewer for pointing this out, and we will add **SPU** to the related work in the revision. We did not include SPU experimentally because we did not have sufficient time to carefully reproduce it under our setting.
>
> **ZSCL** is a full-fine-tuning continual learning method, whose cost is substantially high, so we could not include it in the main results. We additionally evaluate EBO on the MTIL benchmark (Zheng et al., ICCV 2023) using CLIP (Radford et al., ICML 2021) with a ViT-B/16 image encoder (Dosovitskiy et al., arXiv 2020), compare it with Continual-FT (FFT) and ZSCL:
>
> |MTIL|MFN $\uparrow$|FWT $\uparrow$|AVG $\uparrow$|
> |---|---:|---:|---:|
> |Continual-FT|77.3|44.6|55.9|
> |ZSCL|83.6|68.1|75.4|
> |EBO|84.1|68.5|76.5|
>
> EBO achieves consistent gains in this broader setting. Although we were unable to run the full EBLoRA variant within the rebuttal period, this still supports the usefulness of the spectrum-balancing component beyond our main low-rank continual adaptation benchmarks.
>
> ## Q4: Newton Schulz orthogonalization
>
> Yes, Eq. 12 can use Newton Schulz orthogonalization as an alternative whitening step. We tested this variant on UCIT:
>
> |Method|MFN $\uparrow$|MAA $\uparrow$|BWT $\uparrow$|FWT $\uparrow$|
> |---|---:|---:|---:|---:|
> |EBLoRA|72.8|82.9|-2.0|34.6|
> |EBLoRA w/ NSO|71.6|82.7|-3.9|34.3|
>
> Newton Schulz is slightly weaker but faster: about $0.85\times$ the training time of the original EBLoRA and about $1.05\times$ that of standard LoRA, showing a practical efficiency-performance trade-off.

---

> > ### Author Rebuttal · Reviewer_EN9A · 2026-04-02
> >
> > I thank reviewers for their comprehensive answer. Most of my concerns have been addressed , given the listed presentation changes the authors willing to do, I am raising my score.

---

> > > ### Author Response · Authors · 2026-04-03
> > >
> > > We thank the reviewer for the thoughtful follow-up and sincerely appreciate the positive reassessment of our work. We are glad that our clarifications and planned revisions have addressed most of the concerns.

---

### Official Review · Reviewer_Dnij · 2026-03-09

**Soundness:** 3
**Presentation:** 3
**Significance:** 3
**Originality:** 4
**Overall Recommendation:** 5
**Confidence:** 4

**Summary:**

This paper tackles a core issue in continual learning (CL) for large models: catastrophic forgetting when using parameter-efficient methods like LoRA. The authors identify that the singular values of standard LoRA updates are highly imbalanced, with most of the "adaptation energy" crammed into a few dominant components. They argue that this imbalance is a primary culprit for forgetting, as these dominant directions are both disruptive to old tasks and vulnerable to being overwritten by new ones. The proposed method, EBLoRA, factorizes the LoRA update to decouple its magnitude from its direction and then impose structural constraints. The authors force the directional components (U, V) to be orthonormal, effectively balancing the energy spectrum, and also constrain new updates to be orthogonal to important gradient directions from past tasks. The results on UCIT and MLLM-DCL benchmarks shows clear improvements in mitigating forgetting over strong baselines.

**Compliance With Llm Reviewing Policy:**

Affirmed.

**Final Justification:**

This paper offers a fresh and principled perspective on LoRA merging by identifying spectral imbalance as a root cause of interference and enforcing constraints on the Stiefel manifold. The authors' response has clearly addressed my concerns. Therefore, I recommend accepting this paper.

**Key Questions For Authors:**

1. How much does EBLoRA increase training time compared to vanilla LoRA?

2. What is the typical percentage increase in GPU memory usage compared to standard LoRA on your hardware?

3. Did you find that the benefits of EBLoRA are consistent across different ranks (r), or does the optimal rank change compared to standard LoRA?

**Limitations:**

yes

**Strengths And Weaknesses:**

## Strength
1. The core idea of this paper is appealing. Rather than focusing solely on preventing interference, it raises a more fundamental question about which properties of the update itself actually cause it. The concept of "spectral imbalance" offers a fresh and intuitive lens through which to understand why straightforward merging of LoRA adapters often fails.

2. The technical development is also strong. The use of the Stiefel manifold provides a principled way to enforce the desired constraints. The proofs in the appendix justifies that the optimization steps are geometrically sound. The fact that the method can be implemented as a wrapper around standard optimizers like Adam makes it much more likely to be adopted in practice.

3. The experimental evaluation is solid. In particular, the controlled merging experiments in Figure 2 directly test the core hypothesis that balancing the spectrum helps mitigate interference. The ablation studies further clarify the contribution of each component (energy balancing, gradient orthogonality, smart initialization).

4. The paper is clearly written, and the visualizations are well designed and easy to interpret.

## Weakness
1. The computational and memory overhead of the manifold projections needs a more substantial discussion. The paper would benefit greatly from a quantitative analysis.

2. Furthermore, the method's conceptual framework, particularly the Stiefel manifold and the projection steps, is complex. The paper is missing a clear, intuitive illustration or schematic diagram. A figure visualizing the core idea—the factorization of Equation 3, the constraint of U to the manifold orthogonal to past gradients (G), and the projection/retraction process—would make the method easier to understand, especially for a broader audience less familiar with Riemannian geometry.

---

> ### Author Rebuttal · Authors · 2026-03-29
>
> ## W1/Q1/Q2: Training cost and storage cost
>
> We thank the reviewer for this important suggestion. We agree that the computational/storage overhead should be discussed more clearly. Below we compare **LoRA** and **EBLoRA** on the UCIT and MLLM-DCL benchmark:
>
> | Metric | Method | UCIT | MLLM-DCL |
> |---|---|---:|---:|
> | Training speed (s/it) | LoRA | 3.28 | 3.62 |
> | Training speed (s/it) | EBLoRA | 4.20 | 4.51 |
> | Training speed | EBLoRA / LoRA | 1.28 | 1.25 |
> | Memory usage (GB) | LoRA | 32.3 | 35.5 |
> | Memory usage (GB) | EBLoRA | 35.5 | 37.4 |
> | Memory usage | EBLoRA / LoRA | 1.10 | 1.05 |
> | Avg. aggregate performance (%) | LoRA | 37.33 | 34.40 |
> | Avg. aggregate performance (%) | EBLoRA | 47.07 | 42.15 |
> | Avg. aggregate performance | EBLoRA / LoRA | 1.26 | 1.23 |
>
> Here, training speed is measured in seconds per iteration (s/it), i.e., the average wall-clock time of one training step. The extra overhead mainly comes from the additional gradient projection operations required by our constrained optimization in the restricted space. Memory usage reports the average peak GPU memory per card over the 4 GPUs used for training each task.
>
> EBLoRA introduces moderate overhead (about $1.25\times$ training time and $1.05$-$1.10\times$ memory usage that of standard LoRA), while yielding substantially larger gains (about $1.23$-$1.26\times$ that of standard LoRA in average aggregate performance). We will add this discussion in the revision.
>
> We also conducted an additional experiment in which we replaced the **whitening retraction step in Eq. 12**, i.e., the orthogonalization used to map the tentative update back onto the restricted Stiefel manifold, with **Newton Schulz orthogonalization**. The task-by-task results on UCIT are shown below:
>
> | Source \\ Target | ImageNet-R | ArxivQA | VizWiz | IconQA | CLEVR | Flickr30k | Avg. |
> |---|---:|---:|---:|---:|---:|---:|---:|
> | Transfer | — | 48.7 | 33.9 | 25.1 | 19.1 | 44.8 | 34.3 |
> | ImageNet-R | 90.5 | 48.7 | 33.9 | 22.1 | 19.4 | 33.7 | — |
> | ArxivQA | 90.6 | 93.5 | 33.9 | 25.1 | 19.7 | 34.1 | — |
> | VizWiz | 89.9 | 94.1 | 61.5 | 28.2 | 19.2 | 52.3 | — |
> | IconQA | 90.0 | 94.1 | 61.9 | 82.1 | 18.3 | 52.1 | — |
> | CLEVR | 89.9 | 93.6 | 61.2 | 77.1 | 69.4 | 52.1 | — |
> | Flickr30k | 88.4 | 94.0 | 47.4 | 77.5 | 66.2 | 56.3 | 71.6 |
> | Avg. | 89.9 | 86.3 | 50.0 | 52.0 | 35.4 | 46.7 | 60.1 |
>
> Compared with our original whitening-based implementation (Tab. 4f), Newton Schulz orthogonalization yields slightly lower performance overall, but leads to faster training in practice. In our experiments, its training speed is about $0.85\times$ that of the original EBLoRA implementation and about $1.05\times$ that of standard LoRA. This suggests that Newton Schulz is a viable alternative when training speed is preferred.
>
> ## W2: Need for a clearer intuitive illustration of the method
>
> We thank the reviewer for this helpful suggestion. We agree that the current presentation of the Stiefel manifold formulation and the projection/retraction process is mathematically dense.
>
> In the revision, we will add an intuitive schematic figure to visualize the core idea of EBLoRA, including: (1) the factorization in Eq. 3, (2) the constraint that $U_t$ lies on the restricted manifold orthogonal to previously stored subspaces $G$, and (3) the projection/retraction steps used during optimization. We believe such a figure will make the method much easier to follow for a broader audience.
>
> ## Q3: Consistency of EBLoRA across different ranks
>
> We thank the reviewer for this important question.
>
> In our experiments, the benefit of EBLoRA is **not obtained by increasing the rank**, and the optimal rank is essentially the same as standard LoRA. The ranks used in our main experiments are summarized below:
>
> | Method | UCIT | MLLM-DCL |
> |---|---:|---:|
> | LoRA-FT | 16 | 32 |
> | O-LoRA | 96 | 160 |
> | CL-MoE | 96 | 160 |
> | SEFE | 16 | 32 |
> | KeepLoRA | 32 | 64 |
> | EBLoRA (Ours) | 16 | 32 |
>
> As shown above, **EBLoRA uses the same rank as standard LoRA-FT** on both benchmarks ($r=16$ for UCIT and $r=32$ for MLLM-DCL), while still achieving better continual learning performance. This suggests that the gain of EBLoRA comes from **better update geometry and a more balanced spectrum**, rather than from increasing the update rank.
>
> For completeness, the large ranks reported for **O-LoRA** and **CL-MoE** come from their multi-expert design; each expert uses rank 16 or 32, and the total effective rank is the sum across experts. We will clarify this point in the revision.

---

> > ### Author Rebuttal · Reviewer_Dnij · 2026-04-03
> >
> > I thank the authors for their comprehensive response. I will keep my original positive score.

---

> > > ### Author Response · Authors · 2026-04-03
> > >
> > > We thank the reviewer for the encouraging follow-up and sincerely appreciate the continued positive evaluation of our work. We are grateful for the reviewer’s time, careful reading, and consideration.

---

### Official Review · Reviewer_dQWX · 2026-03-12

**Soundness:** 2
**Presentation:** 3
**Significance:** 2
**Originality:** 2
**Overall Recommendation:** 4
**Confidence:** 3

**Summary:**

Summary:
This paper proposes EBLoRA for parameter-efficient continual learning. The key claim is that catastrophic forgetting in low-rank continual adaptation is amplified by spectral imbalance in LoRA updates, where a few dominant singular directions absorb most adaptation energy. Based on this observation, the authors reparameterize task updates as \Delta W_t = s_t U_t V_t^\top, impose orthogonality constraints, and optimize the update on a restricted Stiefel manifold. The paper reports strong results on UCIT and MLLM-DCL, with consistent gains in MFN, BWT, and FWT over LoRA-FT, O-LoRA, SEFE, KeepLoRA, and CL-MoE.

**Compliance With Llm Reviewing Policy:**

Affirmed.

**Final Justification:**

I thank reviewers for their comprehensive answer. Most of my concerns have been addressed , given the listed presentation changes the authors willing to do, I remain my score.

**Key Questions For Authors:**

Are the current experiments sufficient to show that this phenomenon is general, rather than being limited to a specific LoRA configuration or a specific benchmark?

**Limitations:**

The evidential link between the proposed mechanism and the method’s effectiveness is still not fully closed. Although the method achieves performance gains, the paper does not yet convincingly show that these gains primarily come from correcting the specific mechanism of spectral imbalance, rather than from more general optimization benefits such as regularization, parameter constraints, or improved optimization stability. In addition, the experimental scope remains limited. If the authors wish to elevate the conclusion to a mechanistic finding, they should further verify its robustness and generality across a broader range of models, task settings, and hyperparameter conditions. For example, it would strengthen the claim to test more diverse model families, such as ViT, Swin, and ResNet backbones, as well as other parameter-efficient tuning schemes beyond LoRA, including Adapters and Prompt/Prefix tuning. It would also be helpful to validate the conclusion under different rank settings, task orders, continual learning protocols, and datasets.

**Strengths And Weaknesses:**

Strengths:
The paper tackles an important problem, proposes a technically coherent method, and demonstrates strong empirical improvements. The ablation study also suggests that the energy-balancing component is a substantial contributor to the final gains.

Weakness：
1. The paper’s causal claim is stronger than what the evidence supports. The current results are more consistent with correlation or contributory influence than with a rigorous causal conclusion.
2. The link between the proposed mechanism and the method’s effectiveness is not yet fully established. While the paper observes spectral imbalance and shows gains from spectrum balancing, it remains unclear whether the improvement truly comes from correcting the claimed mechanism rather than from more general regularization or optimization effects.
3. The experimental evidence is still insufficient to support the central claim. To justify the use of “cause,” the paper should more systematically rule out alternative explanations and verify the robustness of the conclusion across broader settings.
4. The paper somewhat overclaims its contribution. It appears to simultaneously claim a new phenomenon, a new mechanism, and a new method, but the current evidence is not yet strong enough to fully support all of these claims.

---

> ### Author Rebuttal · Authors · 2026-03-29
>
> We thank the reviewer for the important feedback.
>
> ## W1/W4: On the strength and scope of our claims
>
> We agree that the current wording may sound stronger than the scope of evidence we intend to claim, and we will revise the discussion to distinguish more clearly the **phenomenon**, the **mechanism-level interpretation**, and the **method**.
>
> Our intended claims are not all at the same level. More precisely, we claim: $(1)$ an empirical observation that low-rank updates often exhibit strong spectral imbalance, $(2)$ a mechanism-motivated interpretation that this imbalance is an important contributor to forgetting, and $(3)$ a practical method, EBLoRA, derived from this insight. Thus, our goal is not to claim strict formal causality, but to show that spectral imbalance is a strong and practically important driver.
>
> This evidence also goes beyond simple correlation. As shown in Fig. 2b and Fig. 4, making the spectrum more balanced reduces cross-task interference, balanced updates are more robust to interference, and noise with a more balanced singular-value spectrum causes less degradation.
>
> Consistently, Tab. 3 already shows on UCIT that even EBO, which isolates the energy-balancing component, is highly effective. We observe the same pattern on MLLM-DCL, where even without the orthogonality component, EBO already substantially outperforms LoRA:
>
> | Method | MFN $\uparrow$ | MAA $\uparrow$ | BWT $\uparrow$ | FWT $\uparrow$ |
> |---|---:|---:|---:|---:|
> | LoRA | 54.5 | 61.9 | -11.2 | 32.4 |
> | EBO | 62.1 | 66.2 | -5.9 | 32.9 |
>
> EBO alone is already strong, indicating that the energy-balancing component is itself effective.
>
> To further support the mechanism interpretation, we compare EBO with a $U_t S_t V_t$ variant on UCIT. This variant is obtained from EBO by replacing the scalar $s_t$ with a diagonal matrix $S_t$ and differs from EBO only in allowing the singular values to be learned freely, and thus become imbalanced.
>
> |Method|MFN $\uparrow$|MAA $\uparrow$|BWT $\uparrow$|FWT $\uparrow$|
> |---|---:|---:|---:|---:|
> |EBO|70.2|82.2|-5.1|34.3|
> |$U_t S_t V_t$|68.1|80.5|-7.1|30.6|
>
> Notably, this variant still outperforms standard LoRA, suggesting that spectrum balance is a matter of degree rather than binary. We therefore further examine the rank needed to preserve a given fraction of update energy on ImageNet-R:
>
> | Method | Rank@85% | Rank@90% | Rank@95% |
> |---|---:|---:|---:|
> | LoRA | 4.2 | 5.8 | 8.6 |
> | $U_t S_t V_t$ | 11.0 | 12.2 | 13.6 |
> | EBO/EBLoRA | 16.0 | 16.0 | 16.0 |
>
> These results show a clear progression: LoRA is the most imbalanced, $U_t S_t V_t$ is less imbalanced, and EBO/EBLoRA is fully balanced by construction; performance improves in the same order. We will revise the paper to make this claim hierarchy more explicit and avoid overstating the mechanism evidence.
>
> ## W2: On mechanism vs. generic regularization
>
> We agree that gains could in principle come from generic regularization or optimization effects, so we additionally tested this possibility.
>
> Specifically, we varied the weight decay of standard LoRA on UCIT:
>
> |weight\_decay|MFN $\uparrow$|MAA $\uparrow$|BWT $\uparrow$|FWT $\uparrow$|
> |---|---:|---:|---:|---:|
> |0.00|61.4|76.5|-15.4|26.8|
> |0.05|65.2|74.5|-9.4|31.4|
> |0.10|63.8|74.7|-11.8|31.4|
>
> These results suggest that generic regularization can help, but only modestly and not stably across metrics.
>
> Importantly, EBLoRA and the compared baselines are all trained with the same weight decay ($0.00$) and training regularization settings. Therefore, the gain of EBLoRA is unlikely to be explained simply by stronger generic regularization, and is more consistent with the targeted spectrum-balancing mechanism studied in the paper.
>
> ## W3/Q1/L1: On mechanism evidence and generality
>
> We agree that the current evidence should be interpreted with care: our experiments support a mechanism-aligned interpretation, but do not claim to exhaustively rule out every alternative optimization effect.
>
> To further test generality, we additionally evaluate EBO on the MTIL benchmark (Zheng et al., ICCV 2023) using CLIP (Radford et al., ICML 2021) with a ViT-B/16 image encoder (Dosovitskiy et al., arXiv 2020), and compare it with Continual-FT (FFT) and ZSCL:
>
> |MTIL|MFN $\uparrow$|FWT $\uparrow$|AVG $\uparrow$|
> |---|---:|---:|---:|
> |Continual-FT|77.3|44.6|55.9|
> |ZSCL|83.6|68.1|75.4|
> |EBO|84.1|68.5|76.5|
>
> EBO achieves consistent gains over both Continual-FT and ZSCL in this broader setting. Although we were unable to run the full EBLoRA variant under the rebuttal time limit, this result is still informative: EBO isolates the spectrum-balancing component, and its stable improvement suggests that balancing singular values is itself effective.
>
> We will revise the discussion to clarify this scope more carefully: the current evidence supports the usefulness and generality of the spectrum-balancing principle, and we will further test it in broader settings.

---

> > ### Author Rebuttal · Reviewer_dQWX · 2026-04-03
> >
> > Thanks for the effort, I remain my positive score

---

> > > ### Author Response · Authors · 2026-04-03
> > >
> > > We thank the reviewer for the encouraging follow-up and sincerely appreciate the positive evaluation of our work. We are grateful for the reviewer’s time and consideration.

---

### Official Review · Reviewer_9jtA · 2026-03-12

**Soundness:** 2
**Presentation:** 3
**Significance:** 2
**Originality:** 3
**Overall Recommendation:** 4
**Confidence:** 4

**Summary:**

This paper considers making new tasks update naturally preserve previously learned knowledge in parameter-efficient continual learning. Based on the observation that low-rank adaptation exhibits a highly imbalanced singular spectrum, the paper proposes a novel parameter factorization method to balance the learning of knowledge components. To make the knowledge component basis lie on a manifold feasible set, the paper formulates the optimization problem into restricted Stiefel manifold optimization and thus proposes its corresponding approach, EBLoRA. The experiments and theoretical analysis show the effectiveness of the proposed method.

**Compliance With Llm Reviewing Policy:**

Affirmed.

**Final Justification:**

This paper shows originality in its problem formulation and proposed solution. The authors' rebuttal has clearly addressed my concerns. Therefore, I decided to raise my score.

**Key Questions For Authors:**

1. In EBLoRA, $s_t$ is equal across each rank-one directional component, which forces a balanced spectrum. But this design would restrict the update spectrum and may reduce the natural adaptation representation compared to standard LoRA. Can authors discuss the trade-off between spectral balance and representation flexibility? Having a good explanation would help change the evaluation of the work.

2. Is the imbalanced singular value spectrum caused by LoRA itself or by the fine-tuning optimization process? Clarifying the origin of this would help better justify the proposed design. If extending EBLoRA to general parameter-efficient continual learning methods, will it still work? Having a good explanation would help change the evaluation of the work.

3. How to store previous tasks’ gradients is not explicitly discussed in the paper, where, in line 186, it is mentioned that GPM could be a choice. But different storing methods will cause different costs and capabilities, for example, DualGPM [1] adopts different mechanisms for storing and updating gradient subspaces compared to the original GPM. How does EBLoRA interact with different gradient storage strategies? Does EBLoRA remain stable across them? Having a good explanation would help change the evaluation of the work.

    [1]“Adaptive plasticity improvement for continual learning”, CVPR 2023.

4. Can authors compare the training cost and storage cost of EBLoRA with several baselines?

**Limitations:**

The training cost and storage cost should be evaluated when proposing a new optimization solution.

**Strengths And Weaknesses:**

Strengths:

1. The paper utilizes structured updates, which can balance the learning of each rank-one knowledge component.

2. Instead of directly using SGD optimization, the paper proposes a restricted Stiefel manifold optimization, which can not destroy the orthogonality of knowledge components.

Weaknesses:

1. The paper focuses on how to make new tasks learn in a restricted Stiefel manifold formulation, but for mitigating the forgetting issue, Eq. (4) is a common mechanism to make new and old task subspaces orthogonal. The principled analysis for using this Eq(4) should be more clarified, for example, why using Eq.(4) is better than other orthogonal mechanisms.

2. For $s_t$ in the smooth merging, the paper directly uses the average of all singular values without reference or supportive clarification. It should be clarified why using the average of all singular values is better than using the maximum of the singular values. As the paper mentioned, LoRA has a long-tailed distribution of singular values, which means most small singular values may not be important for the update. Thus, a clear discussion should be given.

3. The training cost and storage cost should be discussed in this paper, since the method involves a novel optimization formulation that involves additional computational steps.

4. The typing mistakes should be corrected. For example, in line 097, “approach coined EBLoRA” should be “approach called EBLoRA”.

---

> ### Author Rebuttal · Authors · 2026-03-29
>
> We thank the reviewer for the constructive feedback.
>
> ## W1: Eq. 4
>
> Our core contribution is not Eq. 4 itself, but the identification and mitigation of spectral imbalance in low-rank updates. As shown in Tab. 3 on UCIT, even without Eq. 4, EBO already substantially improves over LoRA, indicating that the main gain comes from balancing the singular value spectrum rather than orthogonality alone. We observe the same pattern on MLLM-DCL:
>
> | Method | MFN $\uparrow$ | MAA $\uparrow$ | BWT $\uparrow$ | FWT $\uparrow$ |
> |---|---:|---:|---:|---:|
> | LoRA | 54.5 | 61.9 | -11.2 | 32.4 |
> | EBO | 62.1 | 66.2 | -5.9 | 32.9 |
>
> Eq. 4 is thus a complementary interference-reduction mechanism. Compared with InfLoRA (Liang and Li, CVPR 2024) and KeepLoRA (Luo et al., ICLR 2026), our constraint is imposed on $U_t$ without freezing part of the parameterization, preserving a larger feasible space.
>
> ## W2: Mean VS Maximum of $s_t$
>
> Our smoothing follows TA (Ilharco et al., ICLR 2023), where the merged update is
> $$
> \Delta W_{\text{merge}}=\lambda\sum_i \sigma_i U_iV_i^\top.
> $$
> In practice, $\lambda$ is selected by search, so the final energy is jointly determined by $\lambda\sigma_i$ rather than by $\sigma_i$ alone. Therefore, replacing the mean singular value with the maximum mainly changes the scale, and the optimal $\lambda$ should correspondingly be smaller. The two choices are approximately equivalent after tuning $\lambda$.
>
> We verify this on UCIT using the same model-merging setup:
> |$\sigma$|$\lambda$|ImageNet-R|ArxivQA|VizWiz|IconQA|CLEVR|Flickr30k|Avg.|
> |---|---:|---:|---:|---:|---:|---:|---:|---:|
> |$\bar{\sigma}$|2.0|94.7|87.1|24.7|82.2|56.6|53.9|66.5|
> |$\sigma_{max}$|2.0|89.8|86.3|15.3|78.3|53.1|46.3|61.5|
> |$\sigma_{max}$|0.5|94.1|87.2|24.3|81.9|55.9|54.1|66.3|
>
> Empirically, the maximum singular value is about $4.2\times$ the mean in this setting, and the best result under max is indeed obtained with a correspondingly smaller $\lambda$. This supports that mean vs. max is largely a rescaling issue once $\lambda$ is properly tuned.
> ### "LoRA has a long-tailed distribution of singular values, which means most small singular values may not be important for the update. "
> We believe the reviewer’s statement mixes up the role of singular values and directions. A smaller singular value only means that the corresponding $u_iv_i^\top$ has less energy; it does not mean that small singular values are unimportant. Our smoothing keeps all directions $U_tV_t^\top$ unchanged and only reshapes the spectrum.
>
> ## W3/Q4: Training cost
>
> We thank the reviewer for this question. Due to the length limit, please refer to our response to **Reviewer Dnij, W1/Q1/Q2** for the detailed training cost.
>
> ## Q1: Flexibility
>
> EBLoRA does not fundamentally reduce representation flexibility. It constrains the singular-value spectrum, while $U_t$ and $V_t$ remain learnable. In Fig. 3, we compare the first-task performance of LoRA and EBLoRA, showing that EBLoRA remains competitive on newly learned tasks.
>
> Furthermore, We compare the **First** performance of two variants on UCIT:
>
> |Method|ImageNet-R|ArxivQA|VizWiz|IconQA|CLEVR|Flickr30k|Avg.|
> |-|-:|-:|-:|-:|-:|-:|-:|
> |EBO|90.6|94.1|61.0|82.8|67.3|55.9|75.3|
> |$U_t S_t V_t$ variant|90.3|93.6|60.8|82.1|68.1|56.4|75.2|
>
> The $U_t S_t V_t$ variant differs from EBO only in that it allows the learned singular-value spectrum to become imbalanced. Despite this, its average first-task performance is nearly the same as EBO. This suggests that enforcing a balanced spectrum does not inherently limit task fitting.
>
> ## Q2: Origin of imbalance
>
> The imbalanced singular value spectrum is **not specific to LoRA**. We also analyze full fine-tuning on the 8-task benchmark used in TSV-M (Gargiulo et al., CVPR 2025) using an ViT-B-32 visual encoder (Dosovitskiy et al., ICLR 2021). Averaged across the 8 tasks, the rank required to retain most spectral energy remains far below the full rank $768$:
>
> |Method|Rank@85%|Rank@90%|Rank@95%|Rank@100%|
> |---|---:|---:|---:|---:|
> |FFT|136.56|183.95|270.25|768.00|
>
> Thus, the imbalance is not caused only by LoRA parameterization, but appears to be a broader phenomenon of adaptation. We focus on LoRA because it is the dominant setting in recent continual VLM methods. If other parameter-efficient continual learning methods exhibit similar imbalance, the same principle should also apply.
>
> ## Q3: Gradient storage
>
> Appendix C provides implementation details. Our current implementation uses a GPM-style strategy. For Eq. 4, GPM and DualGPM are equivalent at the projection level, since projecting out $M_{l,t}$ is equivalent to projecting onto its orthogonal complement. Thus, given the same old-task subspace, they impose the same orthogonal constraint and feasible directions; the difference is mainly memory/computation trade-off rather than constraint geometry.
>
> ## W4: Typo
>
> We thank the reviewer for pointing this out. We will correct the noted typo and other minor writing issues.

---

> > ### Author Rebuttal · Reviewer_9jtA · 2026-04-04
> >
> > Thanks for authors' efforts in addressing my concerns. I will raise my score accordingly.

---

> > > ### Author Response · Authors · 2026-04-04
> > >
> > > We thank the reviewer for the encouraging follow-up and sincerely appreciate the positive reassessment of our work. We are glad that our clarifications were helpful and addressed most of the concerns.

---

### Decision · Program_Chairs · 2026-04-30

**Decision:**

Accept (regular)

**Comment:**

Reviewers found the paper technically sound and supported by solid empirical results. They viewed it as a useful contribution to continual low-rank adaptation, with a meaningful problem formulation and a well-motivated solution. Reviewer 9jtA highlighted the originality of the formulation and proposed method, and Reviewer Dnij viewed the paper as offering a fresh and principled perspective. The rebuttal addressed most of the concerns raised in the initial reviews and resulted in a clearly more positive final assessment. The remaining issues are mainly about presentation rather than the technical soundness of the work.

Overall, I believe the paper should be accepted. I encourage the authors to incorporate the key rebuttal clarifications into the final version and to present the central claim more carefully.